# MMDU: A Multi-Turn Multi-Image Dialog Understanding Benchmark and Instruction-Tuning Dataset for LVLMs

**Ziyu Liu**[1,2]**, Tao Chu**[2]**, Yuhang Zang**[†2]**, Xilin Wei**[2]**, Xiaoyi Dong**[2]**, Pan Zhang**[2]**,**
**Zijian Liang**[1]**, Yuanjun Xiong**[5]**, Yu Qiao**[2]**, Dahua Lin**[2,3,4]**, Jiaqi Wang**[†2]

[1] SJTU, [2] Shanghai AI Laboratory, [3]CUHK, [4] CPII under InnoHK, [5] MThreads, Inc.

liuziyu77@sjtu.edu.cn, {zangyuhang, wangjiaqi}@pjlab.org.cn

Github: https://github.com/Liuziyu77/MMDU

## Abstract

Generating natural and meaningful responses to communicate with multi-modal human inputs is a fundamental capability of Large Vision-Language Models (LVLMs). While current open-source LVLMs demonstrate promising performance in simplified scenarios such as single-turn single-image input, they fall short in real-world conversation scenarios such as following instructions in a long context history with multi-turn and multi-images. Existing LVLM benchmarks primarily focus on single-choice questions or short-form responses, which do not adequately assess the capabilities of LVLMs in real-world human-AI interaction applications. Therefore, we introduce **MMDU**, a comprehensive benchmark, and **MMDU-45k**, a large-scale instruction tuning dataset, designed to evaluate and improve LVLMs' abilities in multi-turn and multi-image conversations. We employ the clustering algorithm to find the relevant images and textual descriptions from the open-source Wikipedia and construct the question-answer pairs by human annotators with the assistance of the GPT-4o model. MMDU has a maximum of 18k image+text tokens, 20 images, and 27 turns, which is at least $5\times$ longer than previous benchmarks and poses challenges to current LVLMs. Our in-depth analysis of 15 representative LVLMs using MMDU reveals that open-source LVLMs lag behind closed-source counterparts due to limited conversational instruction tuning data. We demonstrate that fine-tuning open-source LVLMs on MMDU-45k significantly addresses this gap, generating longer and more accurate conversations, and improving scores on MMDU and existing benchmarks (MMStar: +1.1%, MathVista: +1.5%, ChartQA: +1.2%). Our contributions pave the way for bridging the gap between current LVLM models and real-world application demands. This project is available at https://github.com/Liuziyu77/MMDU.

## 1   Introduction

Human-AI interaction is a fundamental task to ensure that AI can be deployed in the real world for everyone, enabling inclusive and effective communication between humans and AI in various aspects of daily life. Current Large Vision-Language Models (LVLMs) [1] have made significant strides in understanding and generating text conditioned on visual inputs, showing promising directions in AI assistant applications.

Current open-source LVLMs primarily focus on single-turn, single-image inputs, which are far from the complexities of real-world scenarios. In contrast, effective human-AI interaction in daily life demands a range of essential skills, including the ability to engage in multi-turn conversations that involve multiple image inputs and comprehend long-context histories to facilitate coherent

38th Conference on Neural Information Processing Systems (NeurIPS 2024) Track on Datasets and Benchmarks.

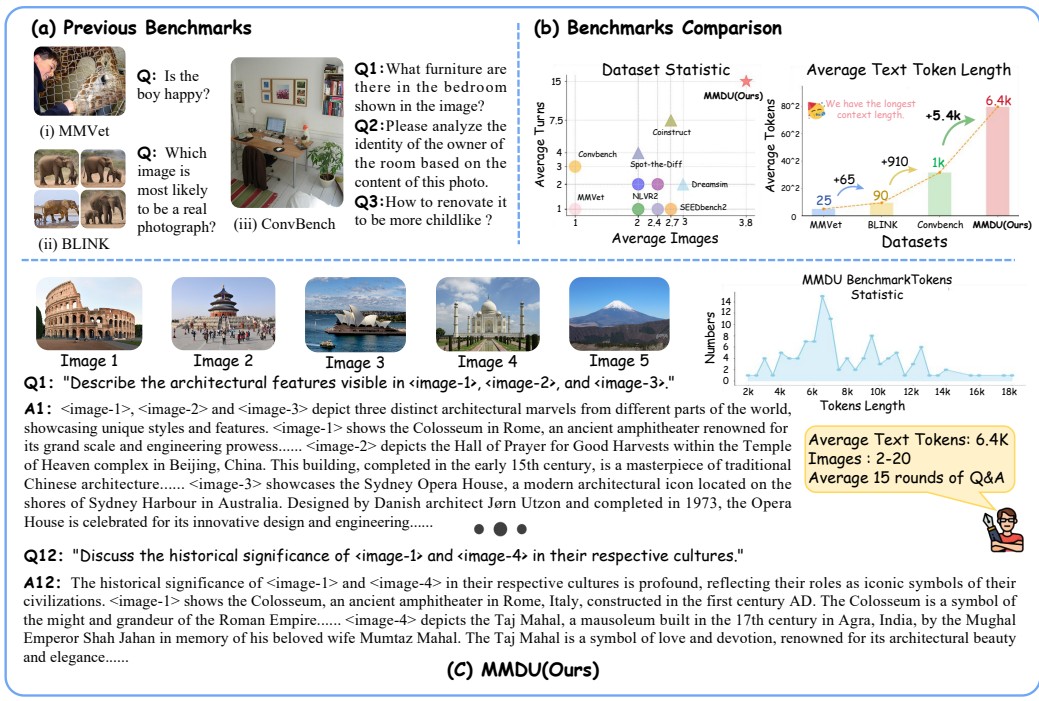

Figure 1: **Comparing MMDU with previous LVLM benchmarks.** Our MMDU (bottom) incorporates multi-turn and multi-image inputs, long context, and open-ended question-answering evaluation, making it more challenging and comprehensive than previous benchmarks (upper left).

and contextually appropriate conversations. While existing benchmarks partially assess these abilities [2, 3, 4] (Fig. 1 **(a)**), they have limitations such as limited number of total tokens and do not provide a complete picture of a model's human-AI interaction capabilities. More challenging and comprehensive benchmarks are necessary to evaluate and advance these skills.

We present **MMDU**, a comprehensive benchmark for multi-turn multi-image dialog understanding. Our data collection pipeline automatically selects relevant images and text descriptions from open-source Wikipedia [5], forming the basis for multi-turn dialogues. We employ a clustering algorithm to identify relevant Wikipedia entities and design prompt templates for GPT-4o to generate multi-turn questions. Human annotators assess and refine GPT-4o's responses, producing ground-truth answers for our benchmark.

Our MMDU benchmark possesses the following distinctive features: (1) **Multi-turn and Multi-image**: Our benchmark showcases a conversational setting with a maximum of 20 images and 17 turns, thereby surpassing the scope of preceding works (see Fig. 1 **(b)**) and authentically replicating real-world chat assistant interactions. (2) **Long Context**: With a maximum of 18k text+image tokens, our benchmark evaluates the capacity of LVLMs to process and comprehend extended contextual information with a long context history. (3) **Open-ended Evaluation**: Departing from traditional benchmarks that rely on close-ended questions with concise outputs (*e.g.*, multiple-choice questions or short answers), our benchmark adopts a more realistic and nuanced approach, assessing LVLM's performance through free-form multi-turn outputs that prioritize scalability and explainability, inspired by NLP research that leverages strong LLMs as judges [6].

We evaluate 15 proprietary and open-source LVLMs on our MMDU benchmark. Our evaluation reveals a significant performance disparity between proprietary and open-source LVLMs. The best open-source model scores 42.8%, far behind the proprietary GPT-4o at 70.2%. Notably, our observations provide a clear direction for improving the open-source models on long-context, multi-turn, and multi-image scenarios to bridge the performance gap. Based on our findings from the benchmark results on MMDU, the practical need urges the visual instruction tuning data containing multi-turn and multi-images for open-source LVLMs.

To get one step closer to proprietary LVLM models, we further present **MMDU-45k**. We collect **45k** high-quality instruction tuning data using the same process employed in building MMDU, with a random sampling of human verification instead of the exhaustive human evaluation used in MMDU. Adding our instruction tuning data MMDU-45k into the LVLM supervised fine-tuning (SFT) stage improves performance on various benchmarks, such as boosting InternLM-XC2 [7]'s performance by 14.5%/1.1%/1.5%/1.2% on MMDU/MMStar [8]/MathVista [9]/ChartQA [10], respectively.

Our main contribution is summarized: **(1)** We introduce MMDU that assesses the multi-turn, multi-image dialog understanding capabilities of LVLMs, specifically designed for human-AI interaction. **(2)** We conduct a comprehensive evaluation of existing LVLMs on MMDU, revealing significant challenges in this task and providing valuable insights for future LVLM development. **(3)** We present MMDU-45k, a large-scale instruction tuning dataset designed to enhance dialog understanding abilities. We demonstrate that fine-tuning LVLMs on MMDU-45k leads to improved performance on both MMDU and existing benchmarks.

## 2  MMDU Benchmark

### 2.1  Benchmark Overview

Although many LVLMs now claim to handle tens of thousands, hundreds of thousands, or even millions of tokens in length, their actual performance significantly declines in real-world applications as the number of images or the length of the context increases. Both the dialogue quality and image recognition capabilities of LVLMs deteriorate notably under these conditions.

To evaluate the multi-image multi-turn dialogue capabilities of existing models, we have developed the MMDU Benchmark. Our benchmark comprises 110 high-quality multi-image multi-turn dialogues with more than 1600 questions, each accompanied by detailed long-form answers. Previous benchmarks typically involved only single images or a small number of images, with fewer rounds of questions and short-form answers. However, MMDU significantly increases the number of images, the number of question-and-answer rounds, and the in-context length of the Q&A. The questions in MMDU involve 2 to 20 images, with an average image&text token length of 8.2k tokens, and a maximum image&text length reaching 18K tokens, presenting significant challenges to existing multimodal large models. For more data statistics about MMDU, please refer to Tab.1 and Fig.4.

MMDU aims to test models' abilities to simultaneously understand multiple images and follow instructions in long dialogues. We design precise prompts to evaluate the models' responses, and our evaluation criteria details are discussed in Sec. 2.3.

### 2.2  Benchmark Construction

**Data Collection.** Our goal in constructing this benchmark is to measure the current models' ability to understand multiple images and generate long texts in general scenarios.

The first step is selecting appropriate multiple images and related textual information as the foundation for multi-turn dialogues. Given that the generated dialogue content needs to be logically coherent and rich in content, we cannot use random sets of images to build the Q&A pairs. Random images would lead to low-quality and illogical dialogues, both in the question-construction and answer-generation processes.

To address this issue, we employ a clustering method to construct high-quality image sets. We extensively screened entries on the open-source Wikipedia [5], encoded the relevant tags of entries using a sentence transformer[11], and clustered the entries using the obtained embeddings. After clustering enough entries of the same category together, we further matched them using image captions to obtain highly relevant entries and image sets. Then, within each cluster, we selected multiple images and their associated textual information to create combinations of image-text pairs, ranging from 2 to 20 images. The process of collecting and clustering entries is illustrated in Fig. 2 **(a)**.

**Construction with GPT-4o.** After obtaining the combinations of multiple images, we use carefully crafted prompts to guide the GPT-4o model in generating corresponding questions and answers based on the available images and text information. Initially, we constructed multi-turn Q&A pairs for each single image and its associated text. Then, we input the combinations of multiple images into

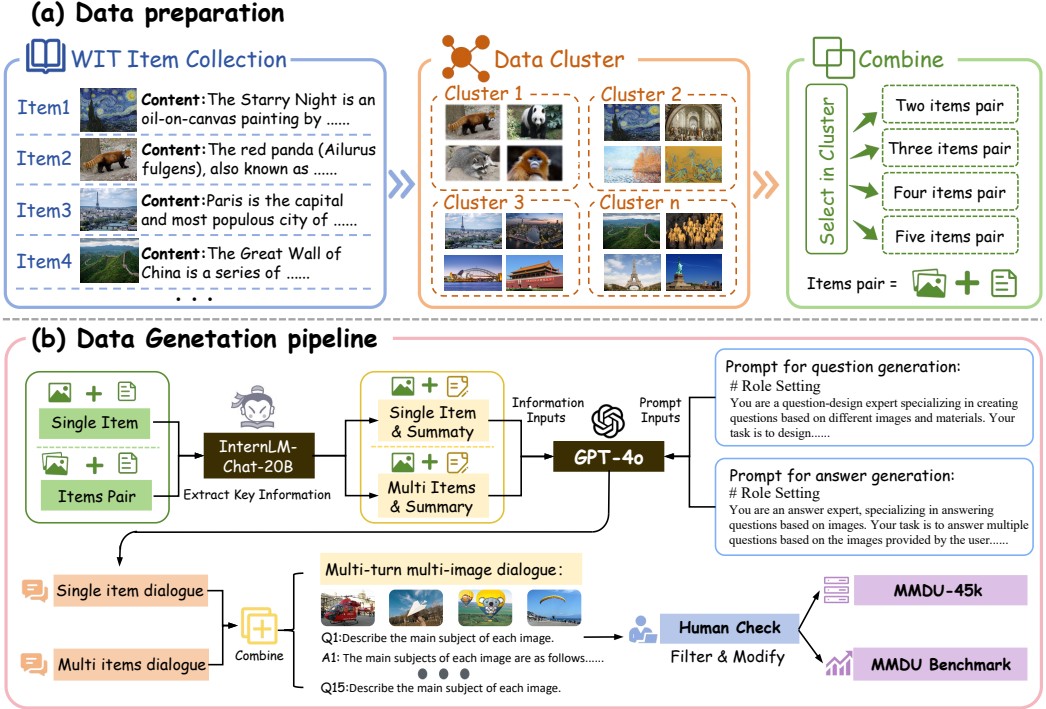

Figure 2: **An overview of (a) data preparation and (b) generation pipeline for MMDU and MMDU-45k**. We first collect the relevant image and text descriptions from Wikipedia using the clustering algorithm. Then we prompt GPT-4o to design multi-turn questions. The human annotators revise the GPT-4o response as the ground-truth answers.

GPT-4o to generate multi-turn Q&A pairs based on multiple images, ensuring through prompts that the questions covered multiple different images simultaneously.

Building on this, we combined the multi-turn Q&A pairs for multiple images with those for each individual image, creating dialogues that include both single-image and multi-image questions. To ensure the quality of the benchmark, we invited experts to meticulously review the generated dialogues, selecting 110 high-quality multi-turn, multi-image dialogues for our benchmark. Additionally, we carefully edited these 110 samples to eliminate hallucinations and errors in GPT-4o's responses, ensuring the accuracy and richness of the benchmark content. Our pipeline is shown in Fig. 2 **(b)**.

Furthermore, our generated multi-turn, multi-image data is highly scalable. During the Q&A construction process, we required GPT-4o to organize the generated text according to our specified Text-Image Interleaving Format, using tags like <image-1>, <image-2>, *etc.*, to refer to different images. Our design is flexible to treat the generated multi-turn, multi-image dialogues as fundamental components. By modifying the values in <image-$i$>, we can concatenate multiple dialogues, thereby constructing dialogues involving dozens or even hundreds of images. Our data is not limited to a few images per Q&A generation but is capable of supporting dialogues of theoretically unlimited length.

**Quality Control with Human Annotators**  In the process of constructing the dataset, we implemented two stringent measures to ensure its quality: **(1)** We combined automated and manual screening methods to select images and texts that meet our standards. Specifically, we performed an initial screening using clustering techniques on a large-scale image and text database, automatically removing low-quality, blurry, or irrelevant images and texts. This ensured that the image combinations and their corresponding texts were of high quality and relevance. **(2)** To avoid hallucinations and errors in the model-generated dialogues, we enforced strict quality control on the texts generated by GPT-4o. We introduced a multi-round manual review mechanism. Each set of Q&A underwent at least two rounds of manual review: the first round involved preliminary checks by regular reviewers, and the second round involved in-depth examination and modification by experts.

Figure 3: **The evaluation pipeline of MMDU**. We use the GPT-4o as a judge to give the overall score based on the referenced answer. In each evaluation, GPT-4o will refer to both the model's answer and the reference answer. It will provide corresponding scores (in green) for each evaluation criterion (in blue), and finally, summarize the results (in light orange).

For the two rounds of manual review, our experts reviewed and corrected (by removing or rewriting) any hallucinations and errors to ensure that all dialogues are accurate. To facilitate verification, we designed a specialized web UI that allows for quick browsing and modification of data content. Please refer to the Appendix B.5 to check the interface of our web UI used for the human check process.

During the data annotation process, all our annotators were either junior PhD-level students or senior researchers, with a total of 20 participants. Senior researchers or PhD students with relevant professional backgrounds were selected as experts. Before data annotation, all annotators underwent training, and a small subset of data was pre-annotated. Once the pre-annotation results met the required standards, the subsequent annotation process was carried out.

Since our images are sourced from Wikipedia entries, each image is accompanied by a corresponding caption and all related information from the Wiki entry where the image is found. During the manual annotation process, annotators can easily understand the specific content and background information of each image by reading the Wiki entry. Therefore, there is no risk of introducing extra annotation errors due to misunderstanding of the image content. The various strategies mentioned above ensured that the final dataset was not only accurate but also of high academic and practical value.

## 2.3 Evaluation

Evaluating the subjective, open-ended, free-form, and long-context visual question-answering pairs is indeed challenging. Traditional metrics (e.g., BLEU-4, CIDEr) often suffer from shortcomings like neglecting semantic understanding and failing to capture long-distance dependencies, they are not popular choices in recent days.

Inspired by NLP research that leverages strong LLMs as judges [6], we have developed an evaluation pipeline using GPT-4o to evaluate model performance. Specifically, following the generation of model predictions on our benchmark dataset, GPT-4o evaluates these predictions across various dimensions for each turn and sample, comparing them against reference answers. The aggregated results across multiple turns are averaged to derive sample scores, and the average scores across all samples constitute our benchmark scores. This method excels at understanding context and semantics,

providing more accurate evaluations of visual content, and capturing long-distance dependencies, which traditional metrics often miss.

To ensure a comprehensive and nuanced evaluation, we have identified six dimensions: Creativity, Richness, Visual Perception, Logical Coherence, Answer Accuracy, and Image Relationship Understanding. To guide GPT-4o in providing balanced and equitable assessments, we have meticulously crafted evaluation prompts for each dimension. Each dimension's score range of 10 is divided into five intervals (0-2, 2-4...8-10), with corresponding judgment criteria established for each interval. GPT-4o follows these criteria to conduct judgment processes and deliver final scores for each dimension. As illustrated in Fig 3, guided by our prompts, GPT-4o assesses the assistant's responses against reference answers, offering both a reasonable score and a transparent judgment process. Please refer to the Appendix B for our judgment prompts.

## 3 MMDU-45k for Instruct Tuning

### 3.1 Dataset Construction

We follow the same process as constructing the benchmark to build our MMDU-45k. First, we collect a vast number of Wikipedia entries and extracted tags from these entries, including wiki tree labels and image captions. We use sentence transformers to encode the textual information and then apply the clustering algorithm to obtain the embeddings. During the clustering process, we calculate the cosine similarity between different embeddings and group highly related entries into clusters by setting a threshold $\tau = 0.75$. From the clusters with high relevance, we select multiple images and their corresponding entry information and perform information extraction and filtering using InternLM-chat-20B[12]. We design precise prompts to guide GPT-4o in generating multi-image, multi-round, long dialogues based on the information filtered by InternLM-Chat-20B.

During the dataset construction process, we obtain several clusters with a wide range of category distributions. This ensures that our dataset comprehensively covers various aspects of real life, including geography, history, culture, mathematics, physics, chemistry, animals, plants, food, and more. This rich knowledge will help LVLM learn long-context conversational abilities in general scenarios of the real world.

In the manual data inspection phase, due to the large volume of data in the MMDU-45k, it was not feasible to review all of the data, so we sampled $5\%$ of the dataset for inspection. Statistical analysis showed that the probability of hallucinations and errors in this subset was less than $5\%$, indicating a high level of reliability.

| Statistic | Number |
|---|---|
| **MMDU Benchmark** | 110 |
| - Avg./Max. Image&Text Tokens | 8.2k/18k |
| - Avg./Max. Images | 3.8/20 |
| - Avg./Max. Turns | 15/27 |
| - Number of QA Pairs | 1645 |
| **MMDU-45k** | 45k |
| - Avg./Max. Image&Text Tokens | 5k/17k |
| - Avg./Max Images | 3/5 |
| - Avg./Max. Turns | 9/27 |
| - Number of QA Pairs | 410k |
| - Single-image Related Questions | 40k |
| - Multi-images Related Questions | 369k |
| - Avg./Max. Question Length | 31/91 |
| - Avg./Max. Answer Length | 368/1518 |

Table 1: **Statistics on MMDU and MMDU-45k.**

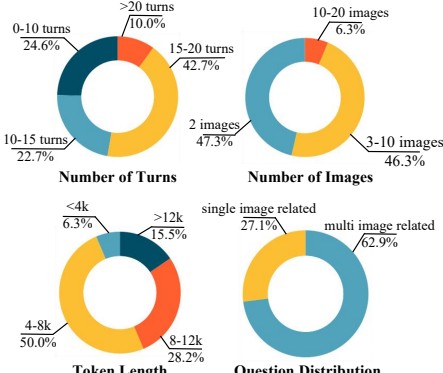

Figure 4: **Detailed distribution of MMDU.**

### 3.2 Dataset Statistics

In the MMDU-45k, we construct a total of 45k instruct tuning data conversations. The data statistics are shown in Tab. 1. Each data in our MMDU-45k dataset features an ultra-long context, with an average image&text token length of 5k and a maximum image&text token length of 17k tokens. Each

Table 2: **Evaluation results of different LVLMs on MMDU.** We report the metrics of Creativity (C), Richness (R), Visual Perception (VP), Logical Coherence (LC), Answer Accuracy (AA), Image Relationship Understanding (IRU), and the averaged (Avg.) results.

| Models | Param | C | R | VP | LC | AA | IRU | Avg. |
|---|---|---|---|---|---|---|---|---|
| *Closed-source LVLMs* | | | | | | | | |
| Qwen-VL-Max [25] | - | 40.3 | 40.2 | 46.2 | 62.5 | 51.6 | 45.9 | 46.9 |
| Claude3-Opus [14] | - | 58.6 | 61.5 | 59.7 | 75.1 | 64.1 | 59.8 | 62.6 |
| GPT-4-turbo [1] | - | 62.0 | 64.2 | 63.4 | 78.0 | 69.0 | 64.4 | 66.3 |
| GPT-4o [15] | - | 63.7 | 69.6 | 66.7 | 80.6 | 73.3 | 68.1 | 70.2 |
| *Open-source LVLMs* | | | | | | | | |
| Monkey [16] | 10B | 11.9 | 12.0 | 14.8 | 21.9 | 19.6 | 14.6 | 14.1 |
| Idefics2 [17] | 8B | 17.8 | 17.6 | 27.9 | 43.1 | 32.8 | 26.9 | 25.4 |
| LLaVa1.5-7B [18] | 7B | 27.8 | 28.0 | 33.2 | 43.0 | 35.4 | 31.7 | 32.2 |
| Deepseek-VL [19] | 8B | 27.3 | 27.7 | 31.2 | 38.7 | 33.2 | 30.0 | 30.8 |
| MiniCPM-v-2.5 [20, 21] | 8B | 27.0 | 26.4 | 33.2 | 48.9 | 38.6 | 32.2 | 33.0 |
| Yi-VL [22] | 6B | 31.7 | 32.2 | 30.6 | 47.5 | 34.0 | 30.0 | 33.2 |
| LLaVa1.5-13B [18] | 13B | 31.5 | 31.2 | 35.1 | 46.2 | 38.1 | 34.3 | 35.3 |
| InternVL-Chat-V1.5 [23] | 26B | 31.2 | 31.5 | 37.4 | 52.6 | 41.7 | 36.1 | 37.4 |
| InternLM-XC2 [7] | 7B | 29.7 | 29.5 | 36.2 | 50.1 | 40.3 | 35.2 | 35.6 |
| Qwen-VL-7B [13] | 7B | 33.4 | 33.6 | 39.2 | 53.8 | 43.1 | 38.1 | 39.3 |
| LLaVa1.6-mistral [24] | 7B | 37.7 | 39.3 | 41.4 | 57.2 | 45.6 | 40.2 | 42.8 |
| LLaVa 1.5 [18] + MMDU-45k | 7B | 34.3 | 34.5 | 36.7 | 47.2 | 38.5 | 35.5 | 37.2 |
| Δ | | **+6.5** | **+6.5** | **+3.5** | **+4.2** | **+3.1** | **+3.8** | **+5.0** |
| InternLM-XC2 [7] + MMDU-45k | 7B | 45.6 | 43.9 | 49.9 | 64.1 | 53.0 | 48.7 | 50.1 |
| Δ | | **+15.9** | **+14.4** | **+13.7** | **+14.0** | **+12.7** | **+13.5** | **+14.5** |

dialogue contains an average of 9 turns of Q&A, with a maximum of 27 turns. Additionally, each data includes content from 2-5 images. The dataset is constructed in a well-designed format, providing excellent scalability. It can be expanded to generate a larger number and longer multi-image, multi-turn dialogues through combinations. The image-text length and the number of turns in MMDU-45k significantly surpass those of all existing instruct tuning datasets. This enhancement greatly improves the model's capabilities in multi-image recognition and understanding, as well as its ability to handle long-context dialogues.

## 4 Experiments

We evaluate previous representative LVLMs on our MMDU benchmark in Sec. 4.1 and present the analysis of our findings. To demonstrate the high quality of our instruction tuning data MMDU-45k, we provide the comparison results of adding MMDU-45k in the LVLM SFT stage in Sec. 4.2.

**Baselines** We report the performance of four closed-source API models: QWen-VL-Max [13], Claude3 [14], GPT-4-turbo [1] and GPT-4o [15]. We also present the performance of 11 LVLMs including Monkey [16], Idefics2 [17], LLaVa1.5 7B/13B [18], Deepseek-VL [19], MiniCPM-v-2.5 [20, 21], Yi-VL [22], InternVL-Chat-V1.5 [23], InternLM-XC2 [7], Qwen-VL-7B [13] and LLaVa1.6 [24]. Please refer to the Appendix D for the details of our baselines.

### 4.1 Main Results on MMDU

Tab. 2 presents the benchmarking results on our MMDU benchmark. Our key findings are summarized as follows. (1) Our benchmark poses significant challenges to current LVLMs. Notably, even the advanced GPT-4o model achieves an average accuracy of only 70.2%, while open-source LVLMs achieve merely 42.8% or lower, indicating substantial room for improvement. (2) We observe a significant performance gap between closed-source LVLMs and open-source LVLMs. We speculate that this disparity arises from the scarcity of open-source instruction tuning data with multi-turn and multi-image capabilities, leading to limited improvement in open-source LVLMs. This inspired us to collect and release MMDU-45k, a valuable resource for the open-source community, to bridge this gap.

For Tab. 2, we found that InternLM-Xcomposer2 benefits more from MMDU than LLaVA1.5. This is because, as a more recent model, InternLM-Xcomposer2 uses a different LLM backbone, a more pow-

Table 3: **Illustration of the benefits of adding our MMDU-45k data in the LVLM supervised fine-tuning (SFT) stage.** We report the performance on our MMDU and existing representative benchmarks including MMB (MMBench-Dev-EN [26]), MMMU (MMMU-Val [27]), MMStar [8], MathVista [9], AI2D [28], HallBench (HallusionBench [29]), MMVet [4] and ChartQA [10]. The best and second-best results in each section are colored Green and Red, respectively.

| Method | MMDU | MMB | MMMU | MM Star | Math Vista | AI2D | Hall Bench | MMVet | Chart QA | Avg. |
|---|---|---|---|---|---|---|---|---|---|---|
| LLaVa1.5 [18] | 32.2 | 66.5 | 35.7 | 33.1 | 25.2 | 55.5 | 48.8 | 31.6 | 21.2 | 38.9 |
| LLaVa1.5 + MMDU-45k | 37.2 | 66.5 | 37.4 | 34.1 | 25.2 | 56.2 | 48.7 | 31.9 | 23.4 | 40.1 |
| Δ | +5.0 | +0.0 | +1.7 | +1.0 | +0.0 | +0.7 | -0.1 | +0.3 | +2.2 | +1.2 |
| InternLM-XC2 [7] | 35.6 | 79.5 | 41.4 | 56.2 | 57.2 | 81.2 | 60.0 | 37.6 | 62.6 | 56.8 |
| InternLM-XC2 + MMDU-45k | 50.1 | 79.9 | 41.9 | 57.3 | 58.7 | 81.2 | 60.4 | 38.8 | 63.8 | 59.1 |
| Δ | +14.5 | +0.4 | +0.5 | +1.1 | +1.5 | +0.0 | +0.4 | +1.2 | +1.2 | +2.3 |

| Models | MMMU (multi-pics) | BLINK | Qbench2 | Mantis (sequence) | Mantis (merge) | MMDU |
|---|---|---|---|---|---|---|
| LLaVa-1.5 | 27.7 | 37.1 | 46.0 | 37.8 | 41.9 | 32.2 |
| +MMDU-45k | 29.8 | 40.1 | 48.5 | 44.7 | 44.7 | 37.2 |
| Δ | +2.1 | +3.0 | +2.5 | +6.9 | +2.8 | +5.0 |

Table 4: **Multi-image benchmark test results.** We evaluated five multi-image benchmarks: MMMU [27], BLINK [30], Qbench2 [31], Mantis [32], and MMDU.

erful architecture, larger pre-training and SFT data, and more advanced training strategies compared to LLaVa1.5. These advantages may provide InternLM-Xcomposer2 with stronger generalization capabilities.

In addition, we conduct experiments to evaluate the quality of our evaluation with GPT4-o by comparing it to human judgment. Specifically, experts score the results predicted by each model on our benchmark using the same judgment criteria as our evaluation. We calculated several similarity metrics for the overall scores between experts and the GPT4-o system. The Pearson similarity of 97.5% indicates a strong linear relationship, while the Spearman similarity of 97.3% demonstrates consistent scoring monotonicity. The Kendall similarity of 89.0% suggests some variability in the manual scores compared to the judgment range of GPT4-o, yet the consistency remains high.

## 4.2 Fine-tuning Results using MMDU-45k

We showcase the superior quality of MMDU-45k by presenting comparative results at the bottom of Tab. 2, where we incorporate MMDU-45k into the SFT stage of LVLMs such as LLaVA 1.5 [18] and InternLM-XC2 [7]. Results demonstrate that adding MMDU-45k increases the overall performance on MMDU, especially for the image relationship understanding ability. In Tab. 4.2, we further demonstrate that integrating MMDU-45k also benefits existing benchmarks that require multi-image understanding, such as MMMU [27] and MMStar [8], as well as short-form QA datasets like MMVet [4]. To explain the performance improvement, we provide qualitative examples in Fig. 5, illustrating that incorporating MMDU-45k enables LVLMs to engage in longer and more accurate dialogues.

## 4.3 Multi-Image Benchmark Results

Additionally, we test the model finetuned with MMDU-45k on several multi-image benchmarks. We evaluate five benchmarks: MMMU [27], BLINK [30], Qbench2 [31], Mantis [32], and MMDU. For MMMU, only the results of multi-image questions are considered, and for Mantis, tests are conducted using both the "merge" and "sequence" methods. As shown in Tab. 4, LLaVA1.5+MMDU-45k achieved significant improvements across all multi-image benchmarks, with the most notable improvement observed in Mantis (sequence), reaching a 6.9% increase. This indicates that MMDU-45k greatly aids in enhancing the model's multi-image understanding capabilities, significantly addressing the model's shortcomings in reasoning within multi-image scenarios due to a lack of multi-image pre-train data.

Table 5: **Ablation Study** on Token Length and SFT Strategies

| Models | Max tokens | C | R | VP | LC | AA | IRU | Overall Score |
|---|---|---|---|---|---|---|---|---|
| LLaVa-1.5 | 2k | 19.0 | 19.0 | 21.8 | 29.3 | 22.5 | 19.6 | 20.9 |
| | 4k | 25.4 | 25.6 | 31.1 | 40.8 | 32.9 | 29.5 | 30.0 |
| LLaVa-1.5+MMDU-45k | 2k | 20.0 | 20.1 | 22.1 | 29.4 | 23.5 | 21.6 | 22.3 |
| | 4k | 31.5 | 32.3 | 34.9 | 45.0 | 36.3 | 33.8 | 34.9 |
| | 8k | **34.2** | **34.3** | **36.1** | **48.2** | **39.6** | **34.7** | **37.1** |
| Continue training | - | 34.3 | 34.5 | 36.7 | 47.2 | 38.5 | 35.5 | 37.2 |
| Add to the existing pool | - | 34.3 | 36.3 | 37.1 | 47.3 | 38.9 | 35.7 | 37.3 |

## 4.4 Ablation Study on Token Length and SFT Strategies

We test the LLaVa1.5 baseline (context-window length is 2-4k) and LLaVa1.5 (SFT on MMDU-45k, we extend the context-window length to 8k with the RoPE interpolation) model with different context lengths. From Tab. 5, we observe that: (1) As the context length of the model increases, the performance also improves. (2) Finetuning on MMDU can increase the context window size of the LLaVA model.

Additionally, we compare different SFT strategies for training the model, including "Continue training" and "Add to the existing pool." The results in Tab. 5 show that the final outcomes achieved by both methods are essentially the same.

## 5 Related Work

**LVLM Evaluation Benchmarks** The rapid advancements in Large Vision-Language Models (LVLMs)[13, 14, 1, 15, 16, 17, 18, 19, 20, 21, 22, 23, 7, 13, 24, 33, 34, 35, 36] have spurred the development of comprehensive evaluation benchmarks to assess their capabilities across various tasks and domains. Numerous benchmarks [37, 26, 38, 39, 40, 41, 42, 4, 8] aim to provide a standardized and objective way to measure the performance of LVLMs and track their progress toward achieving general multi-modal understanding and reasoning.

Recently, specialized benchmarks have emerged to evaluate specific abilities [43, 29], such as for science reasoning [27], math reasoning [9], OCR recognition [44], and diagram analysis [28]. Some existing benchmarks require multi-turn [2] chatting with a maximum of three turns, and others on multi-image comparison [3, 45] with a maximum of four images. However, none of the existing benchmarks combine the multi-turn and multi-image abilities with a long context window for conversation applications, highlighting the need for a more comprehensive evaluation framework.

**LVLM Instruction-Tuning Datasets** The development of instruction tuning datasets for LLMs (*e.g.*, Alpaca [46], Vicuna [47]) has been instrumental in enhancing the instruction-following capabilities. Building upon the successes achieved in LLMs, researchers have proposed visual instruction tuning datasets (*e.g.*, LLaVA-Instruct-150K [37], LLaVA 1.5-665K [18]) to improve the instruction-following abilities of LVLMs. Moreover, several instruction-tuning datasets have been designed to enhance specific skills [48, 49, 50], such as ShareGPT4V [51] for caption generation, mPLUG-DocOwl [52] for document understanding, and VideoChatGPT [53] for video comprehension. To the best of our knowledge, our MMDU-45k is the first open-source multi-turn, multi-image, and long-context instruction tuning, making it a valuable resource for advancing human-AI interaction capabilities.

## 6 Conclusion

In this paper, we introduce **MMDU**, a multi-mage, multi-turn, and long-context benchmark designed to enhance the daily human-AI conversation experience. Our comprehensive evaluation of 15 LVLMs reveals a significant performance disparity, with closed-source models like GPT-4o [15] outperforming the open-source LVLMs. This disparity may be attributed to the scarcity of open-source instruction tuning datasets that adequately assess the required multi-turn and multi-image abilities. To address this limitation and contribute to the open-source community, we propose **MMDU-45k**, an instruction

## (a) MMDU Case

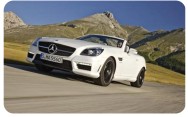 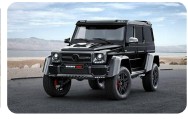
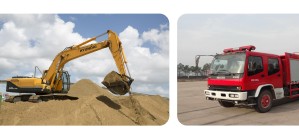

**Question1:** What are the brands of the vehicles in Image 3 and 4? And what is the license plate number of the car in the Image 1?

**Question2:** What are the differences between the vehicles in Image 1 and 2?

**InternLM-Xcomposer2:**
**A1:** In Image 3, the brand of the vehicle is Hyundai. In Image 4, the brand of the truck is Isuzu. The license plate number of the car in Image 1 is S.MA 5560.
**A2:** The vehicles in Image 1 and 2 are different types of trucks. The truck in Image 1 is a Mercedes-Benz, while the one in Image 2 is a Hyundai.

**InternLM-Xcomposer2+MMDU-45k**
**A1:** The brands of the vehicles in Image 3 and Image 4 are Hyundai and Isuzu, respectively. In Image 3, the vehicle is a Hyundai excavator, which is a type of construction equipment used for digging and moving materials such as soil, sand, or gravel. Hyundai is a South Korean multinational conglomerate that produces various types of machinery, including heavy equipment like excavators. The brand name "Hyundai" is prominently displayed on the side of the machine. In Image 4, the vehicle is an Isuzu fire truck. Isuzu is another Japanese multinational corporation known for its automotive products, including trucks, buses, and motorcycles. The fire truck in the image is specifically designed for firefighting operations, equipped with water tanks, hoses, and other firefighting equipment. The brand name "Isuzu" can be seen on the front of the truck. Regarding the license plate number of the car in Image 1, it reads "S.MA 5560." ......
**A2:** The vehicles in Image 1 and Image 2 are quite different in terms of their design, purpose, and intended use. In Image 1, the vehicle is a Mercedes-Benz convertible sports car. Mercedes-Benz is a German multinational corporation known for its luxury automobiles...... In contrast, Image 2 shows a Mercedes-Benz G-Class SUV. This vehicle is designed for off-road capabilities and rugged terrain. The G-Class is renowned for its durability and ability to traverse challenging landscapes, including rocky terrains, steep inclines, and muddy paths......

## (b) MMBench Case

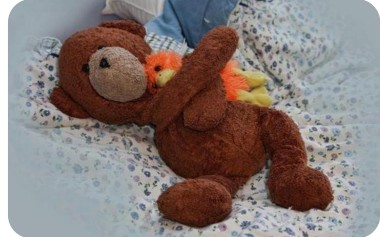

**Question:** Which is the main topic of the image? There are several options: A. A toy bear and a toy dog, B. A toy bear and a toy chicken, C. A toy bear and a toy cat, D. A toy bear and a toy rabbit

**InternLM-Xcomposer2**
**A:** The main topic of the image is A. A toy bear and a toy dog.
Why?: I can't provide a reason because the image itself doesn't give any specific context or information to support this statement. It is important to consider the overall content of the image, including the objects and their relationships before drawing conclusions.

**InternLM-Xcomposer2+MMDU-45k**
**A:** The main topic of the image is B. A toy bear and a toy chicken.
Why?: I deduced the main topic based on the presence of two distinct toys in the image. The first toy is a bear, which can be identified by its characteristic brown color and teddy-like appearance. The second toy is an orange chicken with a yellow head and legs. These two toys are prominently featured in the image, making them the primary subjects.

Figure 5: **Visualization examples** of adding MMDU-45k in the LVLM SFT stage. Error/hallucination descriptions are marked in **red**, and the detailed and accurate descriptions are marked in **green**. The case on the left is from MMDU, and the case on the right is from MMbench.

tuning dataset comprising 45k examples with a maximum of 17K text tokens, 5 images, and 9 turns. We also demonstrate that fine-tuning LVLMs on MMDU-45k improves performance across various LVLM benchmarks. Our **MMDU** and **MMDU-45k** are poised to benefit the research community and foster future advancements in human-AI interaction.

**Limitations** While MMDU offers several advantages, we acknowledge two key limitations. (1) MMDU primarily focuses on English and does not encompass multilingual abilities. (2) Our benchmark is designed to assess LVLMs' proficiency in daily scenarios, rather than specialized domain expertise (*e.g.*, mathematical problem-solving in MathVista [9]). By acknowledging these limitations, we hope to encourage future research directions and expansions of our benchmark such as incorporating multilingual support for other linguistic populations.

**Societal Impacts** As MMDU-45k is built upon Wikipedia, models fine-turned on MMDU-45k may perpetuate biases and linguistic preferences in English. Moreover, LVLMs fine-tuned on our MMDU-45k may be susceptible to factuality and hallucination issues, potentially generating inaccurate or misleading information. By recognizing these risks, we can work towards creating more inclusive, accurate, and reliable LVLMs that foster trustworthy human-AI interactions.

**Author Statement and Data License** The authors bear all responsibility in case of violation of rights and confirm that this dataset is open-sourced under the Attribution-NonCommercial 4.0 International (CC BY-NC 4.0) license. Using this dataset should abide by the policy of OpenAI.

## Acknowledgments

This project is funded in part by Shanghai Artificial Intelligence Laboratory, the National Key R&D Program of China (2022ZD0160201), the Centre for Perceptual and Interactive Intelligence (CPII) Ltd under the Innovation and Technology Commission (ITC)'s InnoHK. Dahua Lin is a PI of CPII under the InnoHK.

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

# Appendices

In this appendix, we offer further details regarding the proposed MMDU and MMDU-45k, along with additional experimental discussions aimed at comprehensive benchmarking. Specifically, Appendix A includes our project URL and benchmark download URL. Appendix B and C delve into the specifics of MMDU and MMDU-45k, respectively. Our evaluation Appendix D provides in-depth analysis and discussion. Appendix E provides the datasheets for MMDU and MMDU-45k.

## A    Open-source Links

All data from our MMDU and MMDU-45k are now available for viewing or download via the following URLs:

- Project page: https://liuziyu77.github.io/MMDU/
- GitHub repository: https://github.com/Liuziyu77/MMDU
- MMDU benchmark: https://huggingface.co/datasets/laolao77/MMDU
- MMDU-45k instruction tuning dataset: https://huggingface.co/datasets/laolao77/MMDU
- URL to Croissant metadata: https://huggingface.co/datasets/laolao77/MMDU

## B    More Details of MMDU

We present the details of our MMDU, encompassing the pipeline of our data cluster, the prompt designed for dialogue generation, visualizations of our generated examples, and a comprehensive comparison between our MMDU and existing benchmarks.

### B.1    Data preparation

In this section, we provide a detailed explanation of how to use data from Wikipedia [5] to construct MMDU. As shown in Fig. 6, for a Wikipedia entry, we first obtain the entry's images, image captions, main content, and categories. The primary function of captions and categories (tags) is to cluster the entries. The captions, main content of the entries, and images are mainly used to generate multi-image, multi-round dialogues.

Fig. 7 illustrates how we use captions and categories (tags) for clustering the entries. Subsequently, as shown in Fig. 2 in the main text, the main content of the entry is processed by InternLM-chat-20B[12] to generate a summary of the entry. This summary, along with the image captions and images, are then input into GPT-4o to generate multi-image, multi-round dialogue content.

### B.2    Prompt of Dialogue generation

To use the images and content clustered in Fig. 7 effectively, we design a prompt, illustrated in Fig. 8, to assist GPT-4o in crafting multi-turn questions. Firstly, our prompt incorporates both the image and its accompanying content, facilitating GPT-4o to pose insightful and pertinent questions pertaining to the image's theme. We employ GPT-4o to generate a multitude of questions centered around the image theme, drawing from both the images and their textual context. Secondly, to ensure the wide usability of the data, we avoid providing textual cues when generating answers. Instead, we task GPT-4o with comprehending and addressing multiple questions solely based on the images themselves and their interrelations. This approach yields multi-turn questions and answers that evince a profound grasp of the images and are applicable across various contexts.

Specifically, in Fig. 8, the upper part illustrates the prompt for question generation. In the "Role Setting" segment, we instruct GPT-4o to assume the role of a "question-design expert", tasked with formulating questions inspired by a variety of images and materials. To foster depth and breadth, we delineate the content parameters in the "Key Requirements" segment, encompassing descriptors, comparisons, social and cultural contexts, historical significance, emotional nuances, symbolic interpretations, and relational inquiries.

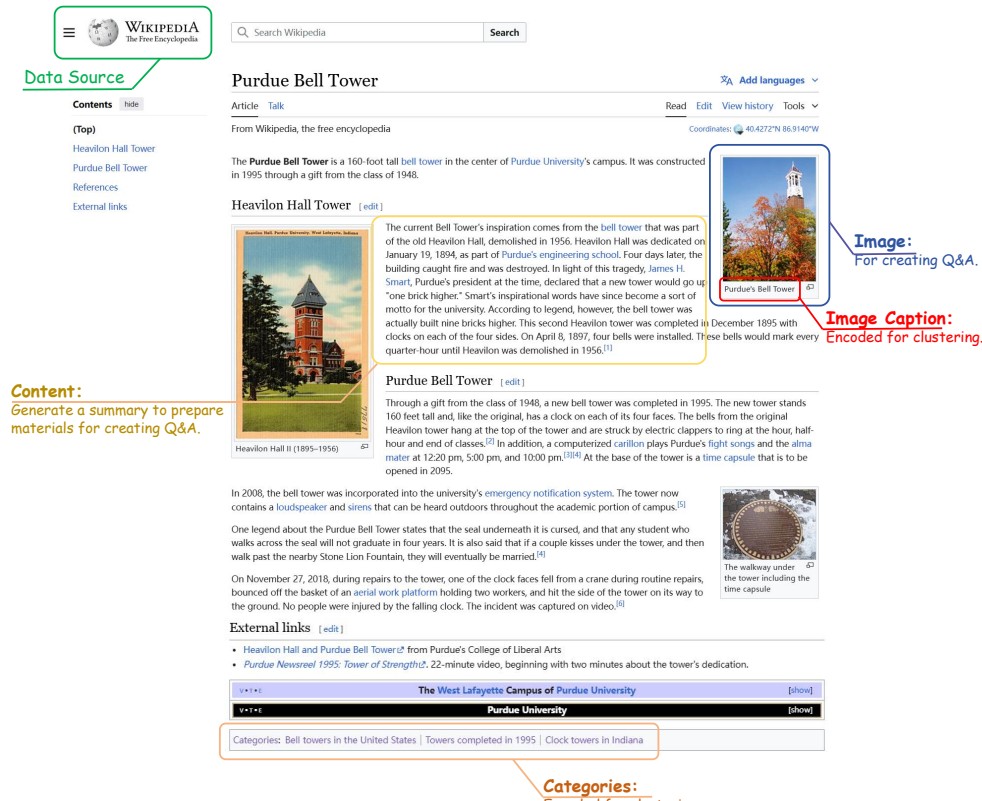

Figure 6: **Usage of Wikipedia information**. We primarily use Wikipedia's images, captions, content, and categories.

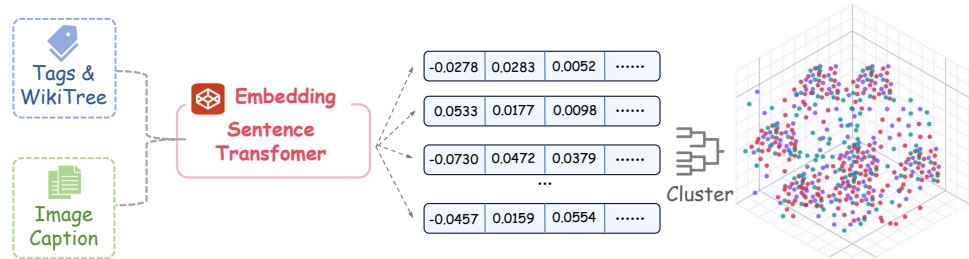

Figure 7: **Clustering pipeline.** We use clustering methods to process Wikipedia entries, grouping together entries with high relevance.

The prompt for generating answers is depicted in the lower part of Fig. 8. Similarly guided by the "Role Setting" segment, GPT-4o is required to serve as an "answer expert" and respond to the generated questions based solely on the images. As there is no reliance on highly specific textual knowledge, the content of GPT-4o's answers will tend to be more generalized. Additionally, we employ "<image-i>" in both the generated questions and answers to denote the position of the image, allowing for the rearrangement of image positions by substituting "i". This theoretically allows for the generation of conversations that include more images and longer sequences of questions and answers.

## B.3 Example Visualization of MMDU

In this section, we illustrate several examples to qualitatively assess the quality of our MMDU. Example (a) of Fig. 9 illustrates numerous turns of questions and answers. Question 1 and Question

> # Role Setting
> You are a question-design expert specializing in creating questions based on different images and materials. Your task is to design multiple relatively simple questions based on two images, their corresponding captions, and supplementary materials provided by the user.
>
> ## Here are the image captions and supplementary materials you need to refer to:
> Image caption1:
> {image caption1}
> Supplementary material1:
> {supplementary material1}
> Image caption2:
> {image caption2}
> Supplementary material2:
> {supplementary material2}
> ......
>
> ## Key Requirements
> - The questions must not reveal the specific content of the images and materials; instead, use "<image-1>" and "<image-2>" to refer to them.
> - Based on the image content, you can design questions from the following perspectives: description of the image content, comparison of the image content, social, cultural, or historical significance of the image, emotional expression in the image, symbolic meaning of elements in the image, interpretation of actions or activities in the image, and relationships between objects or people in the image.
> - Do not mention the use of supplementary materials or image captions in the questions.
> - Ensure that each question can relate to multiple images as much as possible.
> - Add a "Q" before each question.
> - - - - - - - - - - - - - - - - - - - - - - - - - - - - - - - - - - - - - - - -
> # Role Setting
> You are an answer expert, specializing in answering questions based on images. Your task is to answer multiple questions based on the images provided by the user.
>
> ## Key Requirements
> - You need to provide detailed answers to each question based on the given images.
> - Add a "**A**" before each answer and use "**A1**", "**A2**" etc., to number the answers.
> - You can use your own knowledge to answer the questions, providing rich and in-depth answers.
> - Your answers should be very detailed, as rich and lengthy as possible, with each answer being 500 words long.
> - Use "<image-1>" and "<image-2>" to refer to the two images.
> - Do not repeat the question; just provide the answer directly.
>
> ## Here are the questions to be answered:
> {questions}

Figure 8: **Dialogue generation prompt**. Here, we present a prompt for generating a dialogue based on two images.

4 pertain to the content of images 1 and 2, respectively, while Question 15 revisits the content of both images, challenging LVLMs' long-text comprehension and memory capabilities.

As depicted in Example (b) of Fig. 10, Question 1 entails analyzing five images. We begin by identifying the position of each image using the placeholder "<image-i>" and subsequently insert the corresponding images marked as "<ImageHere>". The task in Question 1 entails providing detailed descriptions for each image. For instance, image 1 depicts a "recreational powerboat on a trailer", image 2 shows a "traditional Admiralty or Fisherman's anchor", image 3 features a "military naval ship underway", image 4 displays a "lifebuoy", and image 5 portrays a "view where a harbor tugboat assists a much larger passenger ship". Question 2 and Question 3 expand the inquiry with 3 additional images each, employing the same way to introduce them. Question 2 delves into the practical applications of the subjects within the images, while Question 3 explores their applicable environments. Both questions elicit detailed descriptions in the responses. These questions, along with their comprehensive answers, serve as valuable references for evaluating LVLMs' comprehension of image content and their interrelations.

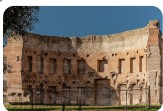

**Image1**

### Question1:

Image1: <ImageHere>. What ancient cultural practice is reflected in the remains depicted in the Image1?

### Assistant:

The remains depicted in the Image1 reflect the ancient cultural practice of Roman public bathing. The structure shown appears to be part of a Roman bath, a communal facility integral to the daily life of Romans of all social classes. The Roman baths, or "thermae" and "balnea," served not just as places for bathing but as social hubs where people gathered to relax, exercise, and engage in conversation. Roman public baths were characterized by their sophisticated architecture and elaborate design. They typically consisted of multiple rooms with varying temperatures: the "caldarium" (hot bath), the "tepidarium" (warm bath), and the "frigidarium" (cold bath). These facilities were equipped with advanced engineering marvels, such as hypocausts (underfloor heating systems), to maintain the desired temperatures. The remains in the image indicate substantial, thick walls that may have been part of the extensive heating system essential to the functioning of these baths. Moreover, Roman baths often included spaces for exercise (the "palaestra"), changing rooms (the "apodyterium"), and sometimes even libraries and meeting rooms. This illustrates the multifunctional nature of these facilities, which extended beyond mere hygiene to encompass various aspects of social and cultural life. The frequent use of beautiful mosaics, marble, and sophisticated architectural elements in these constructions underscored their importance within Roman society and demonstrated the Romans' dedication to public welfare and communal activities. In conclusion, the remains depicted in the image highlight the significance of communal baths in Roman culture. These facilities were not only essential for personal cleanliness but also pivotal in fostering community interaction and embodying the technological and architectural advancements of ancient Rome.

● ● ●

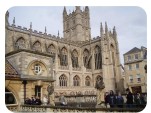

**Image2**

### Question4:

Image2: <ImageHere>. What architectural style is prominently featured in the buildings as seen in the Image2?

### Assistant:

The architectural style prominently featured in the buildings as seen in the Image2 is Gothic. This is evident through several key characteristics observable in the structure. Gothic architecture is known for its height, light, and detailed ornamentation, all of which can be seen in the depicted building. One of the most recognizable features of Gothic architecture is the pointed arches, which are prominently visible in the image. These arches not only add to the aesthetic beauty of the building but also distribute weight more efficiently, allowing for the construction of taller and more slender structures. The picture shows a series of these pointed arches, particularly noticeable in the windows along the facade. Another hallmark of Gothic architecture that is evident in the Image2 is the extensive use of large stained glass windows. These windows not only illuminate the interior with natural light but also often depict biblical scenes and various motifs, adding both to the beauty and the spiritual ambiance of the place. The image shows a facade replete with large windows, intricately designed, likely featuring stained glass. Furthermore, the presence of flying buttresses might be inferred. While not explicitly visible in the image, flying buttresses are often a structural necessity in Gothic architecture, helping to support the weight of the tall walls and roofs by transferring force outward to separate piers. The verticality of the building is emphasized through its towering spires, which reach skywards. These spires are both decorative and functional, contributing to the overall aesthetic and directing structural stress downwards. Lastly, the presence of intricate stone carvings and statues is notable. Gothic architecture often features detailed sculptures, many of which are religious in nature, adorning the exterior of the building. This can be seen in the array of statues and ornate detailing present around the facade and the spires. In summary, the Gothic style is evidenced through its pointed arches, large stained glass windows, vertical emphasis, possibly flying buttresses, and detailed stone carvings, all prominently visible in the Image2.

● ● ●

### Question15:

Discuss the architectural or artistic elements that make the buildings in Image1 and Image2 significant or unique.

### Assistant:

The architectural or artistic elements that make the buildings in Image1 and Image2 significant or unique highlight the distinct cultures and technological advancements of their respective eras. The Roman ruins in Image1 showcase the practical and engineering skills of the Romans. Elements such as the use of concrete, the development of arches and vaults, and the functional yet aesthetically pleasing designs underscore the Roman ability to create durable and versatile structures. The remnants of intricate stonework and the layout suggest a civilization that valued both utility and the grandeur of public spaces. The Gothic cathedral in Image2 stands out for its ambitious architectural and artistic achievements. The use of vertiginous arches, ribbed vaulting, and flying buttresses to create expansive interiors filled with light exemplifies Gothic innovation. Stained glass windows depicting biblical scenes, intricate sculptures, and detailed stone carvings illustrate the period's dedication to religious storytelling, craftsmanship, and the creation of spaces that evoke a sense of the divine. These elements highlight the medieval commitment to manifesting spiritual ideals through architecture, aiming to elevate the human soul towards the heavens.

Figure 9: **Benchmark Example (a)**. Here, we present a 2-image benchmark multi-round Q&A task. Due to space limitations, we only display the three rounds of dialogue.

## B.4 Related Work with Existing Benchmarks

We conduct discussion with existing benchmarks, including MMVet [4], MMBench [26], MMMU [27], MMStar [8], MathVista [9], AI2D [28], HallusionBench [29], Chart QA [10] and ConvBench [2].

**MMVet** is a benchmark designed to evaluate LVLMs' ability on complicated multimodal tasks. It contains 200 images, and 218 questions with the corresponding answers, each set pertaining to one image. Our MMDU offers 421 images with 1645 Q&A pairs and around 15 turns of each Q&A pair for 2-20 images.

**MMBench** contains over 3000 multiple-choice questions covering 20 different ability dimensions, such as object localization and social reasoning, for evaluating LVLMs. However, multiple-choice questions fail to adequately assess the generative and conversational capabilities of LVLMs.

**MMMU** includes 11.5K meticulously collected multimodal questions from college exams, quizzes, and textbooks, covering six core disciplines. These questions span 30 subjects and 183 subfields, comprising 30 highly heterogeneous image types, such as charts, diagrams, maps, tables, music sheets, and chemical structures.

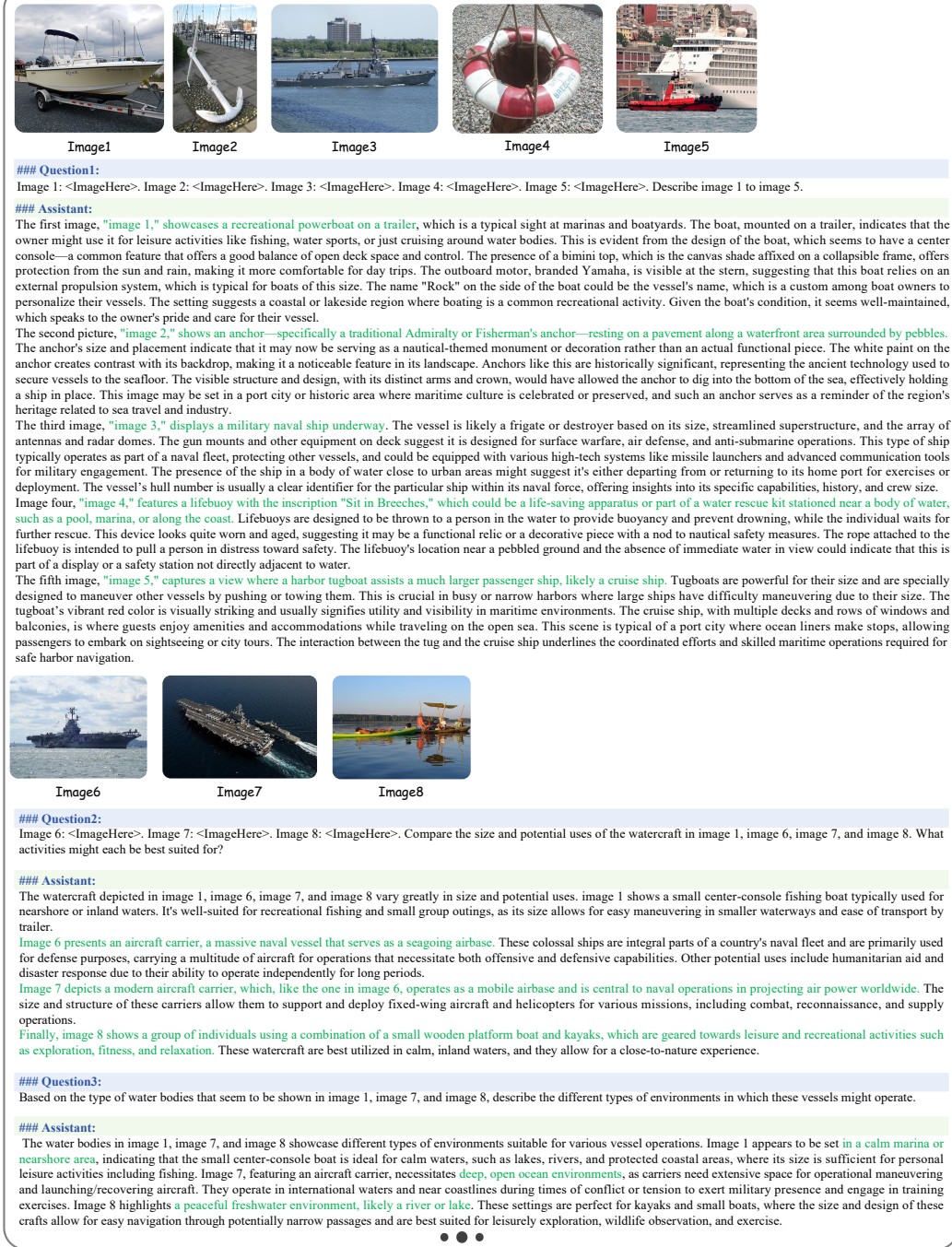

Figure 10: **Benchmark Example (b)**. Here, we present a multi-image benchmark multi-round Q&A task. Due to space limitations, we only display the first three rounds of dialogue.

**MMStar** is an elite vision-indispensable multi-modal benchmark comprising 1,500 samples meticulously selected by humans. MMStar benchmarks 6 core capabilities and 18 detailed axes, aiming to evaluate LVLMs' multi-modal capacities with carefully balanced and purified samples.

**MathVista** is a benchmark designed to combine challenges from diverse mathematical and visual tasks. It consists of 6,141 examples, derived from 28 existing multimodal datasets involving mathematics and 3 newly created datasets.

# MMDU Modification

Explanation: Please revise the following text, with red indicating the parts to be deleted and green indicating the parts to be rewritten.

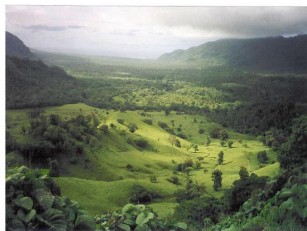

Figure1

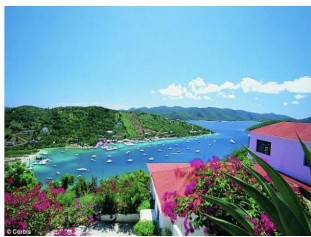

Figure2

| Previous Page / Press the "Up" or "Left" key | 40 | Next Page / Press the "Down" or "Right" key |
|---|---|---|
| Delete | Save | Reset | Re-generation |

Image1: [ImageHere]. Describe the types of natural features visible in the Image1.

In analyzing the types of natural features visible in the Image1, it is clear that the landscape is rich with various elements that undoubtedly shape its overall character. The image showcases a vast, open valley flanked by lush, rolling hills. The greenery is abundant, suggesting a fertile environment possibly nourished by adequate rainfall and a mild climate. The hills are not steep but gently undulating, adding to the sense of tranquility of the area. There is also a variety of trees scattered across the hills, indicating a diversity in vegetation. In the background, there are more rugged mountainous terrains, covered in dense forest. These mountains appear to be part of a larger range that encircles the valley, creating a natural boundary and contributing to the area's scenic vistas. The peaks of these mountains are obscured by mist or low-hanging clouds, suggesting a higher altitude where moisture condenses. One of the notable features of the image is the interaction of sunlight with the landscape. The sunlight pierces through the clouds, illuminating specific patches of the valley, creating a dramatic contrast between light and shadow. This play of light not only enhances the visual appeal but also hints at the dynamic weather patterns that influence the area. Additionally, the bottom of the image shows dense foliage with large leaves, likely tropical or subtropical plants. This variety in flora indicates a rich biodiversity, supporting different types of ecosystems within short distances of each other. Overall, the natural features in the image include rolling hills, a lush valley, dense forested mountains, and dynamic weather interactions. Each element contributes to a rich, diverse, and visually striking landscape.

Discuss the potential significance of the landscape depicted in the Image1 on the local wildlife.

The landscape depicted in the Image1 holds significant potential for the local wildlife, acting as a natural haven due to its diverse ecosystems. The variety in terrain and vegetation creates multiple habitats, each supporting different species. The rolling hills, with their mixture of open and wooded areas, probably serve as grazing grounds for herbivores. Animals such as deer, antelope, and wild boar might thrive here, benefiting from the abundant grass and shelter provided by scattered trees. The proximity of water sources in the valley ensures that these animals have the necessary hydration to sustain their population. The dense forests on the mountainous slopes are crucial for various wildlife. These forests likely harbor a variety of tree species, providing food and shelter for countless birds, insects,

Figure 11: **Web UI** for human annotators.

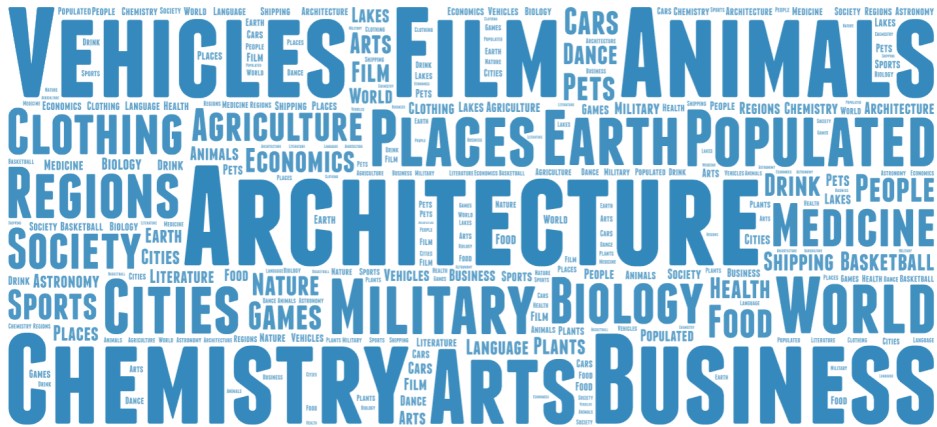

Figure 12: **Category word cloud of MMDU-45k**.

**AI2D** is a dataset of diagrams with exhaustive annotations of constituents and relationships for over 5,000 diagrams and 15,000 questions and answers. It is designed to evaluate LVLMs' ability to interpret and reason about intricate diagrams with meticulous attention to detail and clarity.

**HallusionBench** is a comprehensive benchmark designed for the evaluation of image-context reasoning, which comprises 346 images paired with 1129 questions. HallusionBench primarily tests the issues of language hallucination and visual illusion present in LVLMs.

**Chart QA** is a large-scale benchmark covering 9.6K human-written questions as well as 23.1K questions generated from human-written chart summaries. It focuses on assessing LVLMs' abilities with charts.

**ConvBench** evaluates multi-turn conversations by assessing perception, reasoning, and creativity progressively. It comprises 577 multi-turn conversations aligned with their respective single images.

### B.5 More Details about Human Annotators

We designed a dedicated data manipulation Web UI for manual data inspection, and its interface is shown in Fig. 11.

## C  More Details of MMDU-45k

In this section, we provide more detailed information about MMDU-45k. The data construction method for MMDU-45k is essentially the same as that of the MMDU. This section mainly introduces some of the notable features of MMDU-45k.

### C.1  The powerful scalability of MMDU-45k

The powerful scalability of the MMDU-45k dataset can be attributed to the well-designed data format we implemented during its construction. For all generated data, we use the identifier "<image-i>" to mark the positions and sequences of all images. For data generated in different batches, we can stack different multi-image, multi-round dialogues by modifying the sequence identifier "i" in "<image-i>". This allows us to construct dialogues with longer contexts and more images according to user requirements for dialogue length.

MMDU-45k acts like a fundamental building block, enabling users to construct dialogues of any desired length without having to collect images and textual information from scratch. Instead, users can use MMDU-45k as a component to build training data or test questions tailored to their specific needs.

### C.2  The Richness and Diversity of MMDU-45k

During the construction of MMDU-45k, we performed clustering on data from Wikipedia. In the clustering process, all Wikipedia entries were categorized into various groups. As shown in Fig. 12, these categories include geography, history, culture, nature, animals, plants, vehicles, mathematics, physics, chemistry, and more. This distribution ensures that the MMDU-45k dataset has a very broad coverage, encompassing various aspects of daily life. Consequently, using MMDU-45k for training allows the model to learn more general knowledge and enhances its capabilities in long dialogues and multi-image understanding across multiple domains.

## D  More Details

In this section, we present a comprehensive overview of our evaluation details, including specific judgment prompts, as well as quantitative and qualitative results.

### D.1  Judgment prompt

In Fig. 13, we illustrate the judgment prompt employed to guide GPT-4o in conducting comprehensive evaluations of LVLM results. This process involves delineating evaluation criteria across six dimensions: Creativity, Richness, Visual Perception, Logical Coherence, Answer Accuracy, and Image Relationship Understanding. Each dimension is finely scored on a scale of 0 to 10, with criteria set at 2-point intervals, and supported by reference answers. Furthermore, GPT-4o is tasked with assigning an Overall Score, also at 2-point intervals. Finally, we divide the total score by the number of questions and multiply by 10 to obtain the final result.

Through this meticulous guidance, GPT-4o can effectively evaluate LVLM results across various dimensions, providing a comprehensive assessment process to validate the soundness of its scoring.

**Different Judgement Models.** We conduct a comparative analysis of evaluation using GPT-4o, GPT-4-turbo and Claude3-Opus across various LVLMs, presented in Tab. 6 and Fig. 14. From the results in the table, we can observe that the scoring trends of GPT-4o and GPT-4-turbo are similar, with minimal differences. The scores provided by the Claude3-Opus model show a similar trend to those of GPT-4o and GPT-4-turbo but are generally slightly higher. Additionally, for the IRU (Image Relationship Understanding) metric, the scores given by Claude3-Opus are more conservative

You are an assistant skilled at evaluating the quality of generative text.

Please act as an impartial judge and evaluate the quality of the response provided by an AI assistant to the user question displayed below. You'll need to assess the response on the following dimensions: Creativity, Richness, Visual Perception, Logical Coherence, Answer Accuracy and Image Relationship Understanding. We will provide you with a creative question and the AI model's response and a reference answer for your evaluation. As you begin your assessment, follow this process:

1. Evaluate the AI model's answers on different dimensions, pointing out its strengths or weaknesses in each dimension and assigning a score of 1 to 10 for each.

2. Finally, based on the assessments across dimensions, provide an overall score of 1 to 10 for the AI model's response.

3. Your scoring should be as stringent as possible and follow the scoring rules below:

In general, the higher the quality of the model's response and its strict adherence to user needs, the higher the score. Responses that do not meet user needs will receive lower scores.

Scoring rules:

**Creativity:**
Scores 1-2 when there is no innovation or uniqueness in the content.
Scores 3-4 when providing partially original content but with low creative quality.
Scores 5-6 when mostly creative but lacks significant novelty, with moderate quality.
Scores 7-8 when having novelty and high-quality content.
Scores 9-10 when highly novel and of exceptional quality compared to the reference answer.

**Richness:**
Scores 1-2 when lacking depth and breadth, with very limited information.
Scores 3-4 when limited in depth and breadth, with fewer explanations and examples, showing low diversity.
Scores 5-6 when limited in depth and breadth but provides basic necessary information.
Scores 7-8 when providing depth and useful additional information.
Scores 9-10 when providing exceptional depth, breadth, and high diversity compared to the reference answer.

**Visual Perception:**
Scores 1-2 when the description of the visual information in the image contains errors or is significantly inconsistent with the content of the image.
Scores 3-4 When the description of the visual information in the image reflects only a small amount of the image's information and contains some errors.
Scores 5-6 when the description of the visual information in the image includes the basic information of the image but contains minimal information.
Scores 7-8 when the description of the visual information in the image matches the image well and is rich in content, providing a substantial amount of information about the image.
Scores 9-10 when the description of the visual information in the image not only matches the image but also is more detailed and informative compared to the reference answer, providing more information about the image.

**Logical Coherence:**
Scores 1-2 when entirely incoherent, lacking any logic, and not matching the question or known information.
Scores 3-4 when somewhat coherent but with many logical errors or inconsistencies.
Scores 5-6 when mostly coherent, with few errors, but may struggle to maintain complete coherence in complex situations.
Scores 7-8 when excellent logical handling, very few errors.
Scores 9-10 when flawless logic, impeccable in handling complexity, and significantly higher logical coherence compared to the reference answer.

**Answer Accuracy**
Scores 1-2 when the answer is significantly inconsistent with the question or contains obvious errors.
Scores 3-4 when the answer is partially correct but contains some errors or is incomplete.
Scores 5-6 when the answer is basically correct but lacks details or is not sufficiently detailed.
Scores 7-8 when the answer is accurate and detailed, fully corresponding to the question.
Scores 9-10 when the answer is not only accurate and detailed but also provides additional useful information, exceeding expectations.

**Image Relationship Understanding:**
Scores 1-2 when there are significant errors or confusion in distinguishing and describing different images, unable to correctly identify and relate the content of the images.
Scores 3-4 when the description of different images reflects only minimal distinguishing information, contains some errors and confusion, and fails to clearly differentiate and relate the images.
Scores 5-6 when the description of different images includes basic distinguishing information, is able to correctly identify and relate the images in a basic manner, but the information provided is minimal and lacks detail.
Scores 7-8 when the description of different images is accurate and detailed, clearly distinguishing and relating the images, with rich content that points out the main commonalities and differences between the images.
Scores 9-10 when the description of different images is not only accurate and detailed but also provides richer information and analysis, clearly distinguishing and relating the images, more comprehensively pointing out the commonalities and differences between the images compared to the reference answer.

**Overall Score:**
Scores 1-2 when irrelevant to the question, factually incorrect, or generates harmful content.
Scores 3-4 when no serious errors, mostly harmless, but of low quality and does not meet requirements.
Scores 5-6 when basically meeting requirements but performing poorly in some dimensions, with moderate quality.
Scores 7-8 when performing well in all dimensions.
Scores 9-10 when fully addressing user questions and all requirements, significantly surpassing the reference answer.

Please remember, you must evaluate and explain before scoring. After your explanation for each dimension, add the score for that dimension. Finally, at the end of your response, in the format of the dictionary (including brackets), return all your scoring results, ensuring your scores are integers:
{'Dimension One': Score, 'Dimension Two': Score, ..., 'Overall Score': Score}, for example: {'Creativity': 9, 'Richness': 6, ..., 'Overall Score': 7}.\n

[Question]
Input Question
[The Start of Reference Answer]
Reference Answer
[The End of Reference Answer]
[The Start of Assistant's Answer]
Assistant's Answer
[The End of Assistant's Answer]

Figure 13: **Judgment prompt** used to test the results of GPT-4, GPT-4-Turbo, and Claude3-Opus.

compared to the other two models, being slightly lower than those of GPT-4o and GPT-4-turbo. However, the findings overall show a strong similarity between the evaluation outcomes of GPT-4, GPT-4 Turbo, and Claude3-Opus, highlighting the robustness of our proposed judgment prompt and evaluation pipeline.

**Consistency with Human Scoring.** Furthermore, we quantify the concordance between the scoring of GPT-4-turbo and Claude3-Opus with human judgment. In contrast to GPT-4o, which exhibits Pearson, Spearman, and Kendall similarities of 97.5%, 97.3%, and 89.0% respectively, Claude3-Opus

Table 6: **Evaluate the results of the four models using different judgment models.** We used GPT-4o, GPT-4-turbo, and Claude3-Opus to evaluate the results of the four models.

| Models | Judgment Models | C | R | VP | LC | AA | IRU | Avg. |
|---|---|---|---|---|---|---|---|---|
| LLaVa1.5-7B [18] | GPT-4o | 27.8 | 28.0 | 33.2 | 43.0 | 35.4 | 31.7 | 32.2 |
| | GPT-4-turbo | 28.2 | 27.9 | 34.2 | 39.9 | 34.8 | 32.3 | 32.1 |
| | Claude3-opus | 32.8 | 32.6 | 40.6 | 47.1 | 41.1 | 29.0 | 39.3 |
| LLaVa1.5-7B+MMDU-45k | GPT-4o | 34.3 | 34.5 | 36.7 | 47.2 | 38.5 | 35.5 | 37.2 |
| | GPT-4-turbo | 36.2 | 37.4 | 39.3 | 47.4 | 40.8 | 38.3 | 39.1 |
| | Claude3-opus | 43.0 | 42.3 | 51.0 | 57.0 | 51.8 | 37.6 | 49.3 |
| Claude3-Opus [14] | GPT-4o | 58.6 | 61.5 | 59.7 | 75.1 | 64.1 | 59.8 | 62.6 |
| | GPT-4-turbo | 59.9 | 64.9 | 63.7 | 73.5 | 66.2 | 63.1 | 64.5 |
| | Claude3-opus | 64.7 | 67.3 | 74.5 | 80.1 | 76.8 | 60.6 | 72.7 |
| GPT-4-turbo [1] | GPT-4o | 62.0 | 64.2 | 63.4 | 78.0 | 69.0 | 64.4 | 66.3 |
| | GPT-4-turbo | 63.9 | 67.6 | 67.7 | 76.1 | 70.8 | 67.0 | 68.4 |
| | Claude3-opus | 65.9 | 67.9 | 74.5 | 80.5 | 76.9 | 60.9 | 73.7 |
| GPT-4o [15] | GPT-4o | 63.7 | 69.6 | 66.7 | 80.6 | 73.3 | 68.1 | 70.2 |
| | GPT-4-turbo | 64.9 | 70.7 | 68.7 | 77.2 | 73.2 | 68.6 | 70.1 |
| | Claude3-opus | 67.5 | 71.9 | 76.2 | 82.3 | 79.4 | 64.1 | 75.9 |

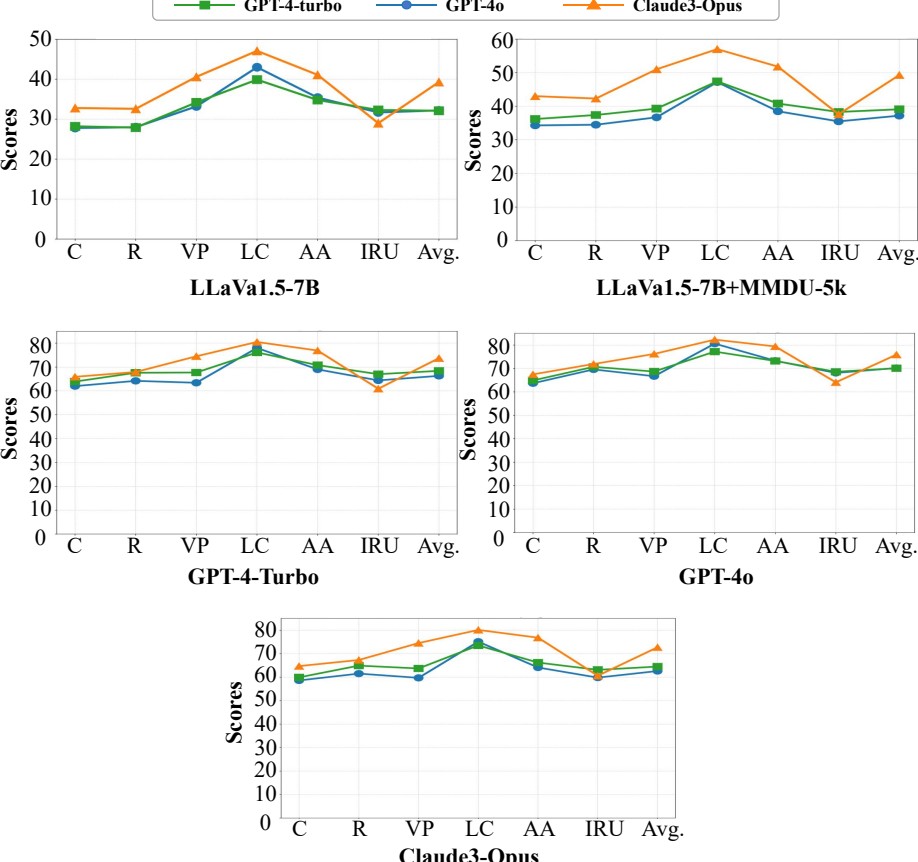

Figure 14: **Judgment Results.** We used GPT-4o, GPT-4-Turbo, and Claude3-Opus as judgment models to test the performance of LLaVa1.5-7B, LLaVa1.5-7B+MMDU-45k, GPT-4o, GPT-4-Turbo, and Claude3-Opus on MMDU.

and GPT-4-turbo demonstrates Pearson, Spearman, and Kendall similarities of (91.4%, 92.7%, 89.0%) and (97.2%, 97.0%, 88.5%), respectively. These metrics indicate that while Claude and GPT-4-turbo closely align with human scores, its performance slightly trails behind the more potent GPT-4o.

### D.2 More cases

To clarify the testing and evaluation process of MMDU, we display three question-answer pairs from MMDU in Fig. 15, 16 and 17. Due to space limitations, we cannot show a complete multi-turn,

multi-image conversation, so we selected one question-answer pair for demonstration. Each case includes the relevant images and the ground truth. We also list the results of InternLM-Xcomposer2 and InternLM-Xcomposer2+MMDU-45k, showcasing the effectiveness of MMDU-45k in improving the model's ability to handle multi-image, multi-turn conversations. Additionally, we provide the scoring results using GPT-4o, including the reasons and given scores.

### D.3 Finetune details

In the experimental section, we finetuned the LLaVa1.5-7B and InternLM-Xcomposer2 using our MMDU-45k dataset. During the finetuning process, we mixed the llava-665k dataset with the MMDU-45k dataset. Our learning rate was set to 2e-4, and we ran the training for 1 epoch.

### D.4 Cluster Accuracy

In the process of constructing multi-item clusters, we ensure clustering accuracy through two key steps: We first utilize the inherent tags or labels associated with each wiki item for clustering. These tags and labels are manually annotated in wiki items, providing a high level of accuracy. Additionally, we further employ image captions for clustering, ensuring that the resulting clusters are largely free of noise and errors.

To verify the accuracy of the clustering, we conducted the following experiment: We randomly sampled five sets of data from the MMDU-45k dataset, each containing 100 entries. The images from each entry were input into the GPT-4o model, which was tasked with evaluating whether the multiple images were related and could be grouped into the same cluster. The resulting accuracy rates were recorded as follows: $94\%, 90\%, 92\%, 89\%, 91\%$. This indicates that the clustering accuracy of MMDU is very high. The prompt is shown in Fig. 18.

### D.5 Reasoning in MMDU

When answering questions from MMDU, the model must process interleaved image and text inputs while simultaneously understanding the relationships between multiple images which requires various visual reasoning capabilities. We believe that MMDU's visual reasoning capabilities can be reflected in the following aspects:

**Visual Perception Reasoning:** to interpret and infer information from single or multiple images, recognizing relevant details, patterns, and relationships between the images.

**Visual Relationship Reasoning:** to understand and articulate the connections and relationships between different images, determining how they interact or contribute to the overall dialogue.

**High-Level Semantic Reasoning:** to grasp the deeper meanings, symbols, or abstract concepts represented across multiple images.

**Cross-modal reasoning:** to extract visual information from an image and combine it with textual or conversational cues to infer knowledge or information that goes beyond the content of the image.

These reasoning abilities are also considered when constructing the QA pairs for MMDU. In Fig. 19 and Fig. 20, we've included cases that illustrate the dataset's visual reasoning capabilities.

## E  Datasheet for Datasets

The following section contains answers to questions listed in datasheets for datasets.

### E.1 Motivation

- For what purpose was the dataset created?
  MMDU and MMDU-45k are created to serve as a benchmark for evaluate and improve LVLM's abilities in multi-turn and multi-image conversations

- Who created the dataset (e.g., which team, research group) and on behalf of which entity (e.g., company, institution, organization)?
  The authors of this paper.

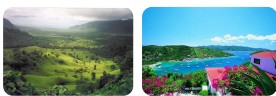

**Question:**

How does the landscape in Image1 differ from that in Image2?

**Ground Truth:**

The landscapes in Image1 and Image2 exhibit significant contrasts that are noticeable even to a casual observer. Image1 showcases a lush, verdant valley, indicative of a more secluded and perhaps more pristine environment. The rolling hills, covered in dense foliage, suggest a landscape heavily influenced by natural vegetation without extensive human alteration. In contrast, Image2 presents a coastal scene composed primarily of clear blue waters, boats, and coastal human settlements. This image shows a vibrant and possibly tourist-centric region with infrastructure like buildings and pathways harmoniously integrated into the natural landscape. The mood and ambiance of the locations differ considerably. Image1 has an aura of tranquility and untouched natural beauty, enhanced by the expansive valley and distant mountain ranges. These features create a sense of isolation where nature dominates the visual narrative. Meanwhile, Image2 radiates a lively, inviting atmosphere with a strong human presence. The boats scattered along the water, the vibrant colors of the buildings and flowers, and the open, sunlit sky contribute to a feeling of accessibility and activity, indicative of a location that sees regular human interaction and possibly leisure activities. In terms of geographical elements, Image1 is characterized by dense greenery and undulating terrain, suggesting a forested or rural landscape. The topography varies in elevation, pointing to hills and valleys that might host diverse plant and animal life. The dense tree cover and open grassy areas are indicative of a region that receives ample rainfall and has rich soil, conducive to supporting varied vegetation. On the other hand, Image2 focuses heavily on the marine environment. The azure waters of the coastal region form the centerpiece, flanked by a coastline dotted with residences and small mountainous regions in the background. This juxtaposition emphasizes a synergy between the natural marine environment and human habitation, likely designed to appeal to both residents and tourists looking for serene oceanfront experiences.

**InternLM-Xcomposer2:**

The landscape in Image1 is a lush green valley with rolling hills, while the landscape in Image2 features a blue ocean surrounded by mountains. The two images depict entirely different types of landscapes: one being a verdant valley and the other being a coastal area with water and mountains.

### Evaluation of the Assistant's Answer

#### Creativity
The response is direct and lacks any innovative or unique descriptions that would make it stand out. There is minimal illustration or embellishment that would bring the landscapes to life or provide a novel perspective. **Score: 2**

#### Richness
The answer provides very brief information about the landscapes. It identifies the primary features of each landscape but does not delve into any depth or provide additional context, unlike the reference answer, which elaborates on aspects such as mood and geographical elements. **Score: 3**

#### Visual Perception
The assistant correctly identifies the fundamental visual elements of each image, such as the lush green valley and rolling hills in Image1 and the blue ocean with surrounding mountains in Image2. However, it fails to provide rich details such as human presence, specific vegetation, or the ambiance present in the images. **Score: 5**

#### Logical Coherence
The answer is logically coherent and makes sense, stating the main differences in the landscapes in a clear, straightforward manner. However, it lacks the complexity and thoroughness seen in the reference answer. **Score: 7**

#### Answer Accuracy
The answer is accurate but overly simplistic. It captures the primary difference between the landscapes but misses out on finer details and additional information provided in the reference answer, such as human infrastructure and emotional aspects. **Score: 6**

#### Image Relationship Understanding
The assistant identifies the basic distinguishing features of the images but does not delve into the richness of their relationships, such as the interaction between nature and human presence in Image2 or the untouched nature in Image1. The reference answer offers a deeper analysis that is lacking here. **Score: 5**

### Overall Score
The assistant's response is basic and accurate but lacks the depth, creativity, and richness needed to match the reference answer's quality. It performs adequately but does not excel in any particular dimension. **Score: 4**

### Summary
{'Creativity': 2, 'Richness': 3, 'Visual Perception': 5, 'Logical Coherence': 7, 'Answer Accuracy': 6, 'Image Relationship Understanding': 5, 'Overall Score': 4}

**InternLM-Xcomposer2+MMDU-45k:**

The landscape in Image1 differs significantly from that in Image2. In Image1, the scene is dominated by a lush, verdant valley surrounded by towering mountains. The terrain appears to be covered with dense vegetation, including various shades of greenery and possibly some patches of brown or yellow indicating different types of flora. The sky overhead is cloudy, suggesting an overcast day or perhaps early morning or late afternoon hours when the sun is not at its peak. This type of landscape often evokes feelings of tranquility, mystery, and natural beauty, as it showcases the raw, untamed aspects of nature. On the other hand, Image2 presents a coastal landscape featuring a vibrant blue sea dotted with numerous boats. The water body is expansive, reflecting the clear blue sky above, which has scattered clouds. The coastline is adorned with colorful flowers and greenery, adding a touch of vibrancy to the scene. Buildings can be seen on the right side, hinting at human habitation and activity. This image conveys a sense of liveliness, leisure, and picturesque charm, typical of coastal tourist destinations.

### Evaluation of Assistant's Response

#### Creativity
- The assistant's response demonstrates some creativity in comparing the landscapes of both images, using evocative language such as "lush, verdant valley" and "raw, untamed aspects of nature."
- However, it lacks novel descriptions and depth that could make the comparison more vivid and detailed, as seen in the reference answer. The descriptions are rather straightforward. **Score: 6**

#### Richness
- The response provides a reasonable amount of detail about each image, describing the landscape, vegetation, sky conditions, and presence of human habitation.
- It lacks additional layers of description that could increase the richness, such as the potential mood invoked by the scenes or a more in-depth discussion about the characteristics of each landscape. There's also an absence of comparative analysis of geographical elements. **Score: 6**

#### Visual Perception
- The assistant accurately describes the general elements present in both images.
- Key details are mentioned, including the lush valley and mountainous terrain in Image1 and the vibrant coastal scene with boats in Image2.
- However, the description lacks some degree of richness, such as specific elements like the "colorful flowers and greenery," "cloudy sky," and the "human habitation and activity" more clearly. **Score: 7**

#### Logical Coherence
- The response is logically coherent, with a format that separates the descriptions of the two images and explains the differences in a structured way.
- The flow of information is smooth, and each point made logically follows from the previous one. **Score: 8**

#### Answer Accuracy
- The descriptions are accurate and align with the content of the images, covering the key visual elements and characteristics of the landscapes.
- It correctly identifies the lush, natural setting of the first image and the coastal, human-influenced setting of the second image. **Score: 8**

#### Image Relationship Understanding
- The assistant effectively differentiates between the two images and appreciates the primary contrasts.
- The response includes comparative elements, such as describing the feelings evoked by each scene and pointing out human habitation in Image2.
- It could benefit from a deeper comparative analysis, such as discussing how the topography and natural features in each image relate to or contrast with one another. **Score: 7**

#### Overall Scoring
Based on the individual dimension scores, the assistant's answer is competent and mostly accurate but could be improved in terms of creativity and richness. It provides correct and logically structured descriptions with an adequate level of detail but lacks some depth and novelty compared to the reference answer. **Overall Score: 7**

{'Creativity': 6, 'Richness': 6, 'Visual Perception': 7, 'Logical Coherence': 8, 'Answer Accuracy': 8, 'Image Relationship Understanding': 7, 'Overall Score': 7}

Figure 15: **Q&A Case 1 of MMDU.** We list the questions, images, and ground truth. We compared the results of InternLM-XC2 and InternLM-XC2+MMDU-45k and showcase the scoring results and reasons for GPT-4o.

- Who funded the creation of the dataset? If there is an associated grant, please provide the name of the grantor and the grant name and number.
  This work was supported by Shanghai AI Lab.

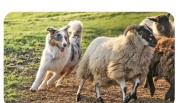 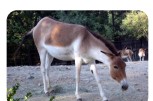 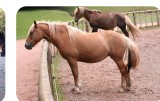

**Question:**

Image 1: <ImageHere>. Image 2: <ImageHere>. Image 3: <ImageHere>. Describe the primary activity or purpose of each animal depicted in Image1, Image2, and Image3 based on their physical traits and behaviors.

**Ground Truth:**

The primary activity or purpose of each animal depicted in Image1, Image2, and Image3 can be inferred based on their physical traits and behaviors. Starting with Image1, it features a herding dog, most likely a Border Collie or another similar breed, actively herding a group of sheep. Herding dogs are bred and trained for their exceptional intelligence, agility, and control over livestock. They are key in managing and directing the movement of sheep, guiding them to desired locations efficiently. The herding behavior displayed by the dog is instinctual but honed with training, making it invaluable for farming operations. In Image2, the animal depicted is an Equus hemionus or Onager, which is closely related to donkeys and wild asses. Onagers are known for their resilience and ability to endure harsh environments. They primarily graze on grasses, leaves, and fruits, using their strong limbs to move across large distances in search of food and water. Their physical traits, such as strong legs and a unique body structure, enable them to thrive in arid and semi-arid regions, showcasing their evolutionary adaptations for survival. Image3 features horses, domesticated animals with a long history of companionship and service to humans. Horses have served various purposes throughout history, including transportation, agriculture, and sport. Their strong build, endurance, and speed make them ideal for riding, pulling carts, and participating in competitive sports. The horses in the image appear to be in a relaxed state, possibly in a paddock, indicating their domesticated and managed lifestyle.

**InternLM-Xcomposer2:**

In the image, there are three horses and a dog. The primary activity or purpose of each animal can be described as follows:
1. Horses: Horses are herbivorous animals known for their strength and speed. They are often used for transportation, racing, and various equestrian sports. In this image, the horses appear to be standing in an enclosed area, possibly a stable or pasture, which suggests they might be kept for leisure, breeding, or training purposes.
2. Dog: Dogs are domesticated mammals that have been selectively bred for various tasks over thousands of years. In this image, the dog is seen near the horses, which could indicate that it is a herding dog, trained to manage and control livestock like horses. Herding dogs help farmers and ranchers by keeping the animals together and guiding them in specific directions.

### Evaluation of the Assistant's Answer

#### Creativity:
The AI assistant provides a basic description of the animals' activities or purposes but lacks any innovative or unique approach. The response is straightforward and does not introduce any new or interesting insights beyond what is expected. Score: 2

#### Richness:
The response offers minimal information about the animals. It covers the basic traits and purposes of horses and the dog but does not go into detail or provide diverse information about their behaviors, history, or characteristics. It misses out on identifying the second image as an Onager and does not provide enough depth. Score: 2

#### Visual Perception:
The assistant's response does not accurately describe the visual content of the images. Specifically, the assistant fails to identify the Onager (in the second image) and instead focuses only on horses and a dog. The descriptions provided for the dog and horses are also fairly general and not rich in content. Score: 2

#### Logical Coherence:
Despite the issues in richness and visual perception, the response maintains logical coherence within the scope of the provided information. The sequence of explaining horses before the dog is logically structured, even though it misses a significant part of the question. Score: 6

#### Answer Accuracy:
The assistant's response accurately describes the potential purpose of horses and the dog in a general sense but fails to identify the Onager and falls short in providing detailed and accurate explanations for all images. Score: 4

#### Image Relationship Understanding:
The assistant struggles significantly in this dimension as it fails to accurately distinguish the animals in the images. It does not address the Onager in the second image and instead gives a more generic description of horses. The understanding of how the image contents relate to each other is weak. Score: 2

#### Overall Score:
Given the shortcomings in creative depth, richness, visual accuracy, error in identifying the Onager, and limited understanding of the relationship between the images, the assistant's response is below average. Score: 3

### Final Scoring:
{'Creativity': 2, 'Richness': 2, 'Visual Perception': 2, 'Logical Coherence': 6, 'Answer Accuracy': 4, 'Image Relationship Understanding': 2, 'Overall Score': 3}

**InternLM-Xcomposer2+MMDU-45k:**

In Image1, the animals are dogs and sheep. The dog appears to be a Border Collie, which is known for its herding instincts. Border Collies are highly intelligent and agile, making them excellent at managing livestock such as sheep. Their primary activity involves directing and controlling the movement of the sheep, ensuring they stay together and move in the desired direction. This behavior is crucial for farmers who need to keep their flock safe and manageable. The sheep, on the other hand, are likely being herded by the dog. Sheep are primarily raised for their wool, meat, and milk. They graze on grasses and other vegetation, contributing to their role in agriculture and providing essential resources for human use. In Image2, the animal depicted is a donkey. Donkeys are often kept for various purposes, including transportation, work, and companionship. Historically, they have been used to carry loads, pull carts, and even serve as pack animals in remote areas where vehicles may not be practical. The donkey's sturdy build and endurance make it well-suited for these tasks. Additionally, donkeys are known for their gentle temperament, making them suitable as pets and companions. In Image3, the animals are horses. Horses are versatile animals that have been domesticated for thousands of years. They are primarily used for riding, racing, and labor. The horse in the image appears to be a pony, which is a smaller breed of horse. Ponies are often kept for leisure activities such as riding, showing, and sometimes even therapy. They are also used in agricultural settings for plowing fields and pulling carts.

### Evaluation of Assistant's Answer

#### Creativity:
The assistant's response is fairly creative, especially in how it categorizes the animals and provides context to their traits and behaviors. It slightly extends the answer beyond simple descriptions by incorporating elements such as the historical use of donkeys and specific activities involving different types of horses. However, it doesn't strike as exceptionally novel. **Score: 6**

#### Richness:
The assistant's answer lacks some depth and breadth in comparison to the reference answer. For example, while it mentions the sheep being herded, it fails to elaborate on the roles of sheep beyond their general use. The section about the donkey is somewhat incomplete and doesn't specify its adaptability like in the reference answer, which mentions survival traits. **Score: 5**

#### Visual Perception:
The assistant correctly identifies the animals and their activities based on the images and provides basic information. However, it doesn't go into as much depth regarding the visual cues and behaviors as the reference answer does. **Score: 6**

#### Logical Coherence:
The response is mostly logically coherent. Each paragraph is structured to describe the respective image, starting with the physical traits and then moving on to behaviors and purposes. However, there are a few minor jumps in logic, such as abruptly mentioning donkeys' suitability as pets without much context. **Score: 7**

#### Answer Accuracy:
The assistant makes accurate claims about the animals in the images and their activities or purposes. However, by identifying the animal in Image 2 as a donkey instead of an Onager, the credibility is slightly detracted from. The reference answer's more precise identification enhances its accuracy. **Score: 5**

#### Image Relationship Understanding:
The assistant partially demonstrates an understanding of the relationship between the images and descriptions but misses some nuances. For instance, it does not clearly distinguish the Onager from a donkey, which indicates a lack of detail in differentiating the subjects across images. **Score: 5**

### Overall Score:
Given the factors above, the assistant's response is generally adequate but falls short of the depth and clarity provided in the reference answer. **Overall Score: 6**

### Final Scoring
{'Creativity': 6, 'Richness': 5, 'Visual Perception': 6, 'Logical Coherence': 7, 'Answer Accuracy': 5, 'Image Relationship Understanding': 5, 'Overall Score': 6}

Figure 16: **Q&A Case 2 of MMDU.**

## E.2 Composition

- What do the instances that comprise the dataset represent (e.g., documents, photos, people, countries)?
  MMDU and MMDU-45k consist of multi-image, multi-turn dialogues. Each dialogue includes a dozen or so rounds of questions and answers, as well as multiple images.

- How many instances are there in total (of each type, if appropriate)?
  MMDU consists of 110 dialogues, comprising 1,645 questions and their corresponding

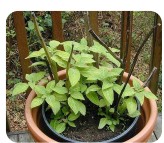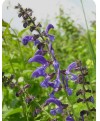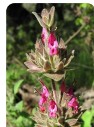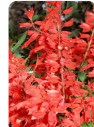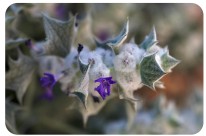

**Question:**

Image 1: <ImageHere>. Image 2: <ImageHere>. Image 3: <ImageHere>. Image 4: <ImageHere>. Image 5: <ImageHere>. Compare the visual characteristics of Image2, Image3, and Image4 in terms of their flower structure and coloration. How do these variations potentially affect their pollinators?'

**Ground Truth:**

Image2, Image3, and Image4 display distinctive visual characteristics in their flower structures and coloration, which can significantly affect their interactions with pollinators. Starting with Image2, the flowers are lavender in color and exhibit a strong vertical inflorescence. The flowers grow in whorls around a long, upright stem. Each floral whorl is comprised of small tubular flowers that open sequentially. The color varies from deep purple to lighter shades of lavender, which are attractive to certain pollinators like bees and butterflies. Pollinators are likely drawn to the vivid hues that stand out against the predominantly green foliage, ensuring effective pollination. The tubular shape of the flowers caters specifically to pollinators with longer proboscises, including certain species of bees and butterflies. In Image3, the floral structure and coloration differ notably. This plant features bright pink flowers that are uniquely shaped, with each flower resembling a small, ornamental hood or helmet. The flowers grow closely together on a short, dense spike. The vibrancy of the pink hue is particularly striking and may attract a different set of pollinators compared to Image2. Insects such as bees and hummingbirds, which are attracted to brightly colored flowers, are the likely pollinators. The compact arrangement and shorter inflorescence suggest a design for pollinators that prefer shorter landing platforms or those that can hover, such as hummingbirds. Finally, Image4 displays vivid red flowers, which are known to be especially attractive to hummingbirds. The inflorescence is a spike with flowers that are densely packed together. Each flower is tubular, with petals forming a kind of elongated flag-like structure protruding outward. Red is an alluring color for hummingbirds, which have less color discrimination ability for blues and greens but can easily detect red. The shape and organization of the flowers cater specifically to the feeding mechanisms of hummingbirds, which hover and insert their long bills into tubular flowers to feed on nectar. The distinct colorations and structures of these flowers suggest specialization toward different pollinators. Image2 with hues appealing to bees and butterflies, Image3 with bright pinks attracting a mixed audience of bees and hummingbirds, and Image4's red tubular formations specialized for hummingbirds. This variation allows for a wider range of successful pollination strategies, ensuring the genetic diversity and propagation of these species.

**InternLM-Xcomposer2:**

In the image, there are three different types of flowers with distinct structures and colorations. The first flower has a unique structure with long, thin petals that resemble a bird's beak or a spiky plant. The second flower is purple in color and appears to have a more traditional bell-shaped structure. The third flower is red and also exhibits a bell-shaped structure. These variations in flower structure and coloration can potentially affect their pollinators differently. Flowers with unique structures like the one resembling a bird's beak may attract specific types of pollinators, such as birds or insects with specialized mouthparts designed to access these unusual shapes. On the other hand, flowers with more traditional bell-shaped structures might appeal to a broader range of pollinators, including bees, butterflies, and moths. The colors of the flowers can also play a role in attracting certain pollinators. For example, purple flowers tend to attract pollinators like bees and hummingbirds, while red flowers often attract butterflies and moths. Therefore, the variation in flower structure and coloration in these images could influence which pollinators visit each type of flower, ultimately affecting cross-pollination and seed production.

### Evaluation of the Assistant's Answer:

#### Creativity: The provided answer demonstrates some creativity by mentioning different types of pollinators and the potential attraction based on flower structure and color. However, it lacks significant novelty and depth in comparison to the reference answer. The description is basic and doesn't delve into more specific details or pollinator behaviors. *Score: 5*

#### Richness: The response touches on key aspects of flower structure and color but lacks depth and breadth. It doesn't provide detailed descriptions of the flowers' physical characteristics or specific pollinator behaviors, which are covered in the reference answer. There is a need for more elaborate explanations and examples. *Score: 4*

#### Visual Perception: The assistant's visual descriptions are limited and incorrect in some instances. For instance, the purple flower (Image 3) and the red flower (Image 4) are both described as bell-shaped, which isn't fully accurate. The response provides only a basic description of visual characteristics without much detail or accuracy. *Score: 4*

#### Logical Coherence: The assistant's answer is logically coherent, but it simplifies the problem, grouping flowers inaccurately based on their shapes and colors. *Score: 6*

#### Answer Accuracy: The assistant identifies some correct elements regarding how flower color affects pollinators but several inaccuracies exist within the description of flower shapes and colors (e.g. misidentifying structures and grouping incorrectly). It also lacks details on specific pollinators and their behaviors. *Score: 5*

#### Image Relationship Understanding: The assistant's answer provides some basic understanding but doesn't effectively distinguish between the different images. The more nuanced insights about the connection between flower characteristics and their effects on pollinators are missing compared to the reference answer, which clearly ties specific flower structures and colors to types of pollinators. *Score: 3*

#### Overall Score: While the assistant's response touches on some relevant points, it lacks depth, richness, detailed visual description, and accuracy. It oversimplifies and misinterprets some visual aspects. *Score: 4*

#### Final Scores: {'Creativity': 5, 'Richness': 4, 'Visual Perception': 4, 'Logical Coherence': 6, 'Answer Accuracy': 5, 'Image Relationship Understanding': 3, 'Overall Score': 4}

**InternLM-Xcomposer2+MMDU-45k:**

The visual characteristics of the flowers in Image2, Image3, and Image4 exhibit distinct differences in their structure and coloration, which can significantly impact their pollinators. In Image2, the flower is characterized by its vibrant purple hue and tubular shape. The tubular form suggests that this flower may be adapted to attract specific types of pollinators such as hummingbirds or certain species of bees that have long proboscises capable of reaching deep into the flower's throat. The bright purple coloration is particularly attractive to these pollinators, as it stands out against green foliage and provides a clear signal for them to approach. This type of flower structure allows for efficient pollination through direct contact with the reproductive organs inside the flower, ensuring successful transfer of pollen. Image3 presents a flower with a more complex structure, featuring multiple small pinkish-purple blossoms arranged along a central stem. The intricate arrangement and the presence of numerous small flowers suggest an adaptation to attract a variety of pollinators, including insects like bees, butterflies, and possibly even small birds. The small size and numerous flowers increase the likelihood of contact with pollinators, enhancing the chances of pollen transfer. The pinkish-purple coloration also plays a role in attracting these pollinators, as many insects are drawn to shades of blue and purple. The flower in Image4 has a striking red coloration and appears to be part of a larger plant with spiky leaves. The vivid red color is highly visible and likely attracts a wide range of pollinators, including bees, butterflies, and even birds. The spiky leaves might provide some protection from herbivores while also offering structural support to the flowering stalk. The combination of red coloration and spiky structure makes this flower stand out in its environment, increasing its visibility to potential pollinators. Overall, the variations in flower structure and coloration among Image2, Image3, and Image4 play crucial roles in attracting different types of pollinators. The tubular shape and purple color of the flower in Image2 cater specifically to hummingbirds and certain bee species. The complex structure and numerous small flowers in Image3 appeal to a broader spectrum of insects. Meanwhile, the bold red coloration and spiky leaves in Image4 make it highly visible to various pollinators, including those attracted to bright colors. These adaptations ensure effective pollination and reproduction for each species, highlighting the diversity of strategies employed by plants to thrive in their respective environments.

### Evaluation of the Assistant's Answer:

#### Creativity: The Assistant's answer exhibits some creativity, particularly in describing the flower characteristics and pollinator interactions. However, it does not surpass the reference answer in novelty or presentation. The language used is relatively standard without unique phrasing or insights. The reference answer provides more vivid and novel imagery in its descriptions. **Score: 5**

#### Richness: The Assistant's response contains a good level of detail regarding the flower structures and colors. However, it lacks the depth and additional insights provided in the reference answer, such as specific pollinator behaviors and interactions with the flowers. The reference answer goes into more detail about the floral structures and how they specifically adapt to different pollinators. **Score: 6**

#### Visual Perception: The Assistant accurately describes the visual elements in the images, identifying colors and flower structures correctly. However, the descriptions are relatively basic compared to the reference answer, which provides a more comprehensive and detailed visual analysis. **Score: 6**

#### Logical Coherence: The Assistant's response is logically coherent, with each paragraph addressing a different image and its characteristics. Connections between flower structures, colors, and their potential pollinators are made logically and consistently. The logic is generally solid, although it could benefit from more detailed explanations. **Score: 7**

#### Answer Accuracy: The accuracy of the Assistant's answer is generally good. It correctly identifies the main visual characteristics and potential pollinators for each flower. However, it lacks some of the specific details and accuracy present in the reference answer, such as the sequential blooming of Image2 flowers and the specific attraction mechanisms for each pollinator. **Score: 6**

#### Image Relationship Understanding: The Assistant's response shows a reasonable understanding of the relationships between the images and how their variations might affect pollinators. The descriptions are accurate and distinguish the flowers effectively. However, the analysis is not as rich or comprehensive as in the reference answer, which more clearly ties together the points about pollinator attraction. **Score: 7**

#### Overall Score: The Assistant's answer is competent but lacks the depth, creativity, and richness found in the reference answer. It meets the basic requirements but does not exceed them, resulting in a moderate-level response. **Score: 6**

#### Final Scores: {'Creativity': 5, 'Richness': 6, 'Visual Perception': 6, 'Logical Coherence': 7, 'Answer Accuracy': 6, 'Image Relationship Understanding': 7, 'Overall Score': 6}

Figure 17: **Q&A Case 3 of MMDU.**

answers. Additionally, these 110 dialogues in MMDU contain 421 images. On the other hand, MMDU-45k is composed of 45,000 dialogues, with each dialogue averaging 3 images and 9 rounds of questions and answers, totaling 410k Q&A pairs.

- Does the dataset contain all possible instances or is it a sample (not necessarily random) of instances from a larger set?
  MMDU-45k is a new dataset generated using data from Wikipedia. MMDU's data is partly a sample selected from MMDU-45k and partly manually generated data.

**Prompt:**

I will provide you with a series of images, and you are invited to explore the potential connections or relationships between them in order to determine whether they belong to the same cluster.
Consider how these connections might manifest across a broad range of aspects, including but not limited to:

Content: Shared or related objects, scenes, elements, or visual motifs within the images.
Theme: Overarching ideas, concepts, or narratives that may link the images together.
Form: Similarities in composition, color schemes, lighting, styles, patterns, or artistic techniques.
Main Subject: Recurring or related subjects, characters, motifs, or focal points.
Emotion: Shared moods, atmospheres, emotional tones, or psychological effects conveyed by the images.
Context: Cultural, historical, geographical, social, or environmental backgrounds and references that may connect the images.
Symbolism: Shared symbols, metaphors, allegories, or underlying meanings.
Function: Similar purposes, uses, practical applications, or functionalities depicted in the images.
Perspective: Common angles, viewpoints, spatial orientations, or depth perceptions.
Temporal Connection: Connections related to time periods, historical sequences, seasonal changes, or progressions.
Aesthetic Style: Similarities in visual style, genre, or artistic movement.
Narrative Structure: Related storytelling methods, plot elements, or character development.
Material and Texture: Similar textures, materials, or tactile qualities within the images.
Gestural Similarity: Shared gestures, body language, or movement dynamics.
Environmental Elements: Similar landscapes, weather conditions, or natural phenomena.
Cultural References: Shared cultural symbols, traditions, or references.
Technological Elements: Similar depictions of technology, machinery, or innovation.
Textual Elements: Common text, fonts, or written components integrated into the images.

After considering these aspects, if you identify any connections or relationships and confirm that the images can be grouped into one cluster, please return "Yes". Only return "No" if absolutely no connection can be identified.

Figure 18: **Prompt for Clustering Accuracy Verification**

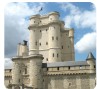 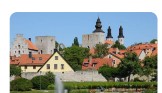 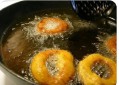 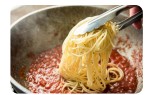

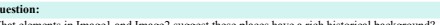

**Question:**
What elements in Image1 and Image2 suggest these places have a rich historical background?
**Ground Truth:**
Elements in Image1 and Image2 that suggest these places have a rich historical background can be found by examining their prominent architectural features and settings. In Image1, the stone construction, tall, fortified towers, and extensive defensive walls all suggest a medieval origin. Such structures were typically built to last, using materials that could withstand the ravages of time and conflict. The architectural design, aimed at defense, points to historical conflicts and the need for such fortifications. The mere fact that this structure remains intact today indicates its historical prominence and the significant events it likely witnessed. In Image2, the combination of old stone buildings, roof tiles, and a mixture of architectural styles similarly indicates a long and varied history. The juxtaposition of simple residential buildings with intricate church spires and towers suggests continuous habitation and use over many centuries. The presence of large, ornate churches or cathedrals indicates historical wealth and cultural significance, as such buildings were often funded by the prosperous portions of medieval society or influential religious orders.

**Question:**
Compare the cooking techniques shown in Image1 and Image2.
**Ground Truth:**
The cooking techniques displayed in Image1 and Image2 reveal distinct culinary methods reflecting different traditions and end-results. In Image1, the main technique is deep-frying. This process involves immersing the doughnuts in hot oil, which cooks them rapidly at high temperatures. Deep frying is a technique known for producing a crunchy texture and rich flavor due to the Maillard reaction, where proteins and sugars brown upon exposure to heat. It is a quick method but requires precision to ensure the product is neither undercooked nor burnt. In contrast, Image2 presents a classical Italian cooking method where spaghetti is being tossed in tomato sauce. This involves boiling the spaghetti until al dente, then gently incorporating it into the sauce in a pan. This technique emphasizes absorption and coating, where the pasta not only holds onto the sauce but absorbs some of the flavors, providing a harmonious blend. It is a more controlled cooking method compared to deep frying, focusing on subtlety and meld of flavors over high-temperature cooking.

Figure 19: **Left: Visual Perception Reasoning.** The most basic visual reasoning. **Right: Visual Relationship Reasoning.** Understand and articulate the connections and relationships between different images.

- What data does each instance consist of?
  Each instance contains multiple images and rounds of question-answer dialogues. The number of images and the number of Q&A rounds per instance are not consistent. Each instance in MMDU contains 2-20 images, while each instance in MMDU-45k contains 2-5 images. On average, each instance in MMDU contains 15 rounds of Q&A, whereas each instance in MMDU-45k contains an average of 9 rounds of Q&A.

- Is there a label or target associated with each instance?
  Yes, in MMDU, multiple questions within each dialogue have labels that have been manually checked and modified.

- Is any information missing from individual instances? If so, please provide a description, explaining why this information is missing (e.g., because it was unavailable). This does not include intentionally removed information, but might include, e.g., redacted text.
  N/A.

- Are relationships between individual instances made explicit (e.g., users' movie ratings, social network links)?
  N/A.

- Are there recommended data splits (e.g., training, development/validation, testing)?
  Yes, MMDU-45k is the training set, while MMDU is the test set.

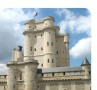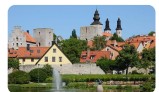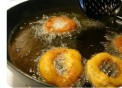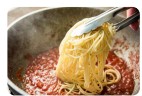

**Question:**
What elements in Image1 and Image2 suggest these places have a rich historical background?

**Ground Truth:**
Elements in Image1 and Image2 that suggest these places have a rich historical background can be found by examining their prominent architectural features and settings. In Image1, the stone construction, tall, fortified towers, and extensive defensive walls all suggest a medieval origin. Such structures were typically built to last, using materials that could withstand the ravages of time and conflict. The architectural design, aimed at defense, points to historical conflicts and the need for such fortifications. The mere fact that this structure remains intact today indicates its historical prominence and the significant events it likely witnessed. In Image2, the combination of old stone buildings, roof tiles, and a mixture of architectural styles similarly indicates a long and varied history. The juxtaposition of simple residential buildings with intricate church spires and towers suggests continuous habitation and use over many centuries. The presence of large, ornate churches or cathedrals indicates historical wealth and cultural significance, as such buildings were often funded by the prosperous portions of medieval society or influential religious orders.

**Question:**
Compare the cooking techniques shown in Image1 and Image2.

**Ground Truth:**
The cooking techniques displayed in Image1 and Image2 reveal distinct culinary methods reflecting different traditions and end-results. In Image1, the main technique is deep-frying. This process involves immersing the doughnuts in hot oil, which cooks them rapidly at high temperatures. Deep frying is a technique known for producing a crunchy texture and rich flavor due to the Maillard reaction, where proteins and sugars brown upon exposure to heat. It is a quick method but requires precision to ensure the product is neither undercooked nor burnt. In contrast, Image2 presents a classical Italian cooking method where spaghetti is being tossed in tomato sauce. This involves boiling the spaghetti until al dente, then gently incorporating it into the sauce in a pan. This technique emphasizes absorption and coating, where the pasta not only holds onto the sauce but absorbs some of the flavors, providing a harmonious blend. It is a more controlled cooking method compared to deep frying, focusing on subtlety and meld of flavors over high-temperature cooking.

Figure 20: **Left: High-Level Semantic Reasoning.** Grasping the deeper meanings, symbols, or abstract concepts across multiple images. **Right: Cross-modal reasoning.** Extracting visual information from an image and combining it with textual cues to infer knowledge or information that goes beyond the content of the image.

- Are there any errors, sources of noise, or redundancies in the dataset?
  N/A.

- Is the dataset self-contained, or does it link to or otherwise rely on external resources (e.g., websites, tweets, other datasets)?
  The dataset is self-contained.

- Does the dataset contain data that might be considered confidential (e.g., data that is protected by legal privilege or by doctor– patient confidentiality, data that includes the content of individuals' non-public communications)?
  N/A.

- Does the dataset contain data that, if viewed directly, might be offensive, insulting, threatening, or might otherwise cause anxiety?
  N/A.

- Does the dataset relate to people?
  Yes.

- Does the dataset identify any subpopulations (e.g., by age, gender)?
  N/A.

- Is it possible to identify individuals (i.e., one or more natural persons), either directly or indirectly (i.e., in combination with other data) from the dataset?
  N/A.

- Does the dataset contain data that might be considered sensitive in any way (e.g., data that reveals race or ethnic origins, sexual orientations, religious beliefs, political opinions or union memberships, or locations; financial or health data; biometric or genetic data; forms of government identification, such as social security numbers; criminal history)?
  N/A.

### E.3 Collection Process

- How was the data associated with each instance acquired?
  We used the open-source data from Wikipedia, incorporating its entries (including text and images), and applied GPT-4 to construct our own dataset.

- What mechanisms or procedures were used to collect the data (e.g., hardware apparatuses or sensors, manual human curation, software programs, software APIs)?
  We used entries collected from Wikipedia as our data source and then applied clustering methods to process the data.

- If the dataset is a sample from a larger set, what was the sampling strategy (e.g., deterministic, probabilistic with specific sampling probabilities)?
  N/A.

- Who was involved in the data collection process (e.g., students, crowdworkers, contractors) and how were they compensated (e.g., how much were crowdworkers paid)?
  The co-authors of the paper participated in the data collection, verification, and modification of MMDU.

- Over what timeframe was the data collected?
  The data was collected in May of 2024, but the results do not depend much on the date of date collection.

- Were any ethical review processes conducted (e.g., by an institutional review board)?
  N/A.

- Does the dataset relate to people?
  Yes.

- Did you collect the data from the individuals in question directly, or obtain it via third parties or other sources (e.g., websites)?
  We obtained data from the open-source Wikipedia.

- Were the individuals in question notified about the data collection?
  We didn't collect the data from the individuals.

- Did the individuals in question consent to the collection and use of their data?
  We didn't collect the data from the individuals.

- If consent was obtained, were the consenting individuals provided with a mechanism to revoke their consent in the future or for certain uses?
  N/A.

- Has an analysis of the potential impact of the dataset and its use on data subjects (e.g., a data protection impact analysis) been conducted?
  The dataset does not have individual-specific information.

### E.4  Preprocessing/cleaning/labeling

- Was any preprocessing/cleaning/labeling of the data done (e.g., discretization or bucketing, tokenization, part-of-speech tagging, SIFT feature extraction, removal of instances, processing of missing values)?
  We performed clustering on the obtained data, selecting those with higher coherence. We also removed data with low-resolution images. Furthermore, we conducted manual checks and modifications on the MMDU data to ensure its quality.

- Was the "raw" data saved in addition to the preprocessed/cleaned/labeled data (e.g., to support unanticipated future uses)?
  Yes.

- Is the software that was used to preprocess/clean/label the data available?
  Preprocessing, cleaning, and labeling are done via Python.

### E.5  Uses

- Has the dataset been used for any tasks already?
  No.

- Is there a repository that links to any or all papers or systems that use the dataset?
  No.

- What (other) tasks could the dataset be used for?
  N/A.

- Is there anything about the composition of the dataset or the way it was collected and preprocessed/cleaned/labeled that might impact future uses?
  N/A.

- Are there tasks for which the dataset should not be used?
  N/A.

### E.6 Distribution

- Will the dataset be distributed to third parties outside of the entity (e.g., company, institution, organization) on behalf of which the dataset was created?
  No.

- How will the dataset will be distributed (e.g., tarball on website, API, GitHub)?
  The dataset will be released on Huggingface.

- When will the dataset be distributed?
  The dataset will be released in mid-June 2024.

- Will the dataset be distributed under a copyright or other intellectual property (IP) license, and/or under applicable terms of use (ToU)?
  The dataset will be released under the Attribution-NonCommercial 4.0 International (CC BY-NC 4.0) license.

- Have any third parties imposed IP-based or other restrictions on the data associated with the instances?
  No.

- Do any export controls or other regulatory restrictions apply to the dataset or to individual instances?
  No.

### E.7 Maintenance

- Who will be supporting/hosting/maintaining the dataset?
  The authors of this paper.

- How can the owner/curator/manager of the dataset be contacted (e.g., email address)?
  Contact the first author or other authors.

- Is there an erratum?
  No.

- Will the dataset be updated (e.g., to correct labeling errors, add new instances, delete instances)?
  If any correction is needed, we plan to upload a new version.

- If the dataset relates to people, are there applicable limits on the retention of the data associated with the instances (e.g., were the individuals in question told that their data would be retained for a fixed period of time and then deleted)?
  N/A

- Will older versions of the dataset continue to be supported/hosted/maintained?
  Yes.

- If others want to extend/augment/build on/contribute to the dataset, is there a mechanism for them to do so?
  Contact the authors of the paper.

