# OpenReview forum: "MMDU: A Multi-Turn Multi-Image Dialog Understanding Benchmark and Instruction-Tuning Dataset for LVLMs"
_NeurIPS.cc/2024/Datasets_and_Benchmarks_Track — NeurIPS 2024 Track Datasets and Benchmarks Poster_

### Official Review · Reviewer_Ckq6 · 2024-07-25
**MMDU: A Multi-Turn Multi-Image Dialog Understanding Benchmark and Instruction-Tuning Dataset for LVLMs**

**Rating:** 8
**Confidence:** 4
**Clarity:** Yes, with minor language mistakes.

**Review:**

This work aims to advance the research in the field of LVLM from single image based QA to multi-turn multi-image dialog understanding by constructing new datasets.
This important effort is of great significance to the field.

**Strengths:**

1) constructed MMDU dataset with human annotation and validation
2) undertook comprehensive evaluation on 15 models on the newly established dataset.
3) constructed MMDU-45K for instruction tuning.

**Additional Feedback:**

N/A

**Correctness:**

Yes, the construction of the dataset is reasonable, though non GPT-4o based method could be used for manual assess of dataset quality.

**Documentation:**

Yes, Github repository has been setup for this project.

**Limitations:**

The discussions on Limitations are reasonable.

**Opportunities For Improvement:**

1) MMDU has only 110 dialogues, although having 1600+ Q&A pairs.
2) More details are needed on the manual annotation process. For example, how are normal reviewers and experts are chosen? How about their knowledge to fully understand the context of images, as this will affect the quality of final results.
3) I am suspect of using GPT-4o to assess the quality of answers.
4) some grammar mistakes were spotted. For example, “significantly address(es)” in Line 19.

**Relation To Prior Work:**

Good and sufficient.

**Summary And Contributions:**

This work presents new benchmark datasets for multi-turn multi-image dialog understanding (MMDU), while existing multi-modal understanding methods focus on single image single turn Q&A.

---

> ### Author Rebuttal · Authors · 2024-08-16
>
> Dear reviewer,
>
> We sincerely thank you for your thorough feedback:  advance the research in the field of LVLM, great significance to the field. We address the questions below and will incorporate all feedback in the final version.
>
> > Q1: MMDU has only 110 dialogues, although having 1600+ Q&A pairs.
>
> Thanks for your comments! MMDU is a multi-image, multi-turn, and long context dialogue benchmark, typically having longer ground truths and a higher number of turns than previous benchmarks.
> On average, each entry in the MMDU Benchmark includes 15 dialogue turns, which means that although our benchmark contains only 110 dialogues, it encompasses 1,600 Q&A pairs.
> Our multi-turn design reflects the model's ability to perform multi-image reasoning in long-context scenarios, posing a significant challenge to existing LVLM models.
>
> > Q2-1: More details are needed on the manual annotation process.
>
> Thanks for your feedback! For the dialogues in the MMDU benchmark, our experts reviewed and corrected (by removing or rewriting) any hallucinations and errors to ensure that all dialogues are accurate. Due to the large volume of data in the MMDU-45k, it was not feasible to review all of the data, so we sampled 5% of the dataset for inspection. Statistical analysis showed that the probability of hallucinations and errors in this subset was less than 5%, indicating a high level of reliability. To facilitate verification, we also designed a specialized web UI that allows for quick browsing and modification of data content. Please refer to the PDF in the General Response section to check the interface of our web UI used for the human check process.
>
> > Q2-2: For example, how are normal reviewers and experts are chosen?
>
> During the data annotation process, all our annotators were either junior PhD-level students or senior researchers, with a total of 20 participants. Senior researchers or PhD students with relevant professional backgrounds were selected as experts. Before data annotation, all annotators underwent training, and a small subset of data was pre-annotated. Once the pre-annotation results met the required standards, the subsequent annotation process was carried out. We will update the details of the manual annotation process in our revised version.
>
> > Q2-3: How about their knowledge to fully understand the context of images, as this will affect the quality of final results.
>
> Since our images are sourced from Wikipedia entries, each image is accompanied by a corresponding caption and all related information from the Wiki entry where the image is found. During the manual annotation process, annotators can easily understand the specific content and background information of each image by reading the Wiki entry. Therefore, there is no risk of introducing extra annotation errors due to misunderstanding of the image content.
>
> >  Q3: I am suspect of using GPT-4o to assess the quality of answers.
>
> Good question! Evaluating the subjective, open-ended,  free-form, and long-context visual question-answering pairs is indeed challenging. In general, the community considers two options:
>
> (1) Traditional metrics (e.g., BLEU-4, CIDEr). Since these metrics often suffer from shortcomings like neglecting semantic understanding and failing to capture long-distance dependencies, they are not popular choices in recent days.
>
> (2) LLM/LVLM models as a judge. This method excels at understanding context and semantics, providing more accurate evaluations of visual content, and capturing long-distance dependencies, which traditional metrics often miss.
>
> While using LLM/LVLM models like GPT-4o as a judge indeed has certain limitations (e.g., sensitivity to evaluation prompts), they are increasingly recognized as better alternatives for evaluation. Recent LLM benchmarks (e.g., MT-Bench [1]) and Multi-modal benchmarks (e.g., MM-Vet [2]) also use the LLM/LVLM for judging.
>
> Besides, we also present the following two proofs to demonstrate the effectiveness of GPT-4o judgment:
> - In line 202 of the main paper, we compared the assessment results of GPT-4o with the human experts. The GPT-4o exhibits Pearson, Spearman, and Kendall similarities of 97.5%, 97.3%, and 89.0% respectively, with human judgment results.
> - In Table 1 and Figure 8 of the Supplementary Material, we compare the difference of using different judgment models (e.g., GPT-4o, GPT-4-turbo, Claude 3-Opus). We find a strong similarity between the evaluation outcomes of different models, and overall the GPT-4o model has the the maximum correlations with the human experts.
> Based on the above-mentioned reasons, we select the GPT-4o as the judgment model.
>
> [1] Judging LLM-as-a-Judge with MT-Bench and Chatbot Arena, Neurips 2023 Datasets and Benchmarks Track
>
> [2] MM-Vet: Evaluating Large Multimodal Models for Integrated Capabilities, ICML 2024
>
> > Q4: some grammar mistakes were spotted. For example, “significantly address(es)” in Line 19.
>
> We apologize for the grammatical errors and appreciate your patience in pointing them out. We will make the necessary corrections to the revised version of the paper.

---

> ### Author Response · Authors · 2024-08-27
>
> Dear Reviewer Ckq6,
>
> We deeply appreciate your efforts in reviewing our paper, and we are particularly grateful for the invaluable suggestions that have greatly enhanced its quality.
>
> In response to your suggestions, we explained the relationship between the number of dialogues and the number of Q&A pairs in Q1. In Q2, we provided a detailed description of the human check process. In Q3, we explained the reasons why using GPT-4o for scoring and compared it with existing subjective evaluation methods.
>
> Could you please read our rebuttal and consider adjusting the score in light of our response? Thank you for your time and effort!
>
> Best,
>
> Authors

---

> > ### Comment · Reviewer_Ckq6 · 2024-08-30
> >
> > Dear Authors,
> > Thanks for the rebuttal. I prefer to keep the original rating.

---

> ### Author Response · Authors · 2024-08-31
>
> Dear Reviewer Ckq6,
>
> We sincerely thank you for your positive feedback on our work. We are delighted to learn that our rebuttal has effectively addressed your concerns regarding (1) the number of dialogue and Q&A pairs, (2) the manual annotation process, and (3) the scoring methodology. We greatly appreciate the time and effort you have dedicated to providing such thorough reviews and insightful comments.
>
> With best regards,
>
> The Authors

---

### Official Review · Reviewer_URmW · 2024-07-29

**Rating:** 6
**Confidence:** 4
**Correctness:** No

**Review:**

The MMDU dataset contains distinct characteristics that make it stand out in the current landscape of existing vision-language benchmarks i.e., larger context length, higher avg. number of images in each sample etc. The evaluation dataset has undergone sufficient quality control. The training dataset roughly follows the same collection process as the evaluation benchmark, so it's no surprise that it significantly improves performance on the MMDU benchmark. The empirical results contain extensive results from multiple contemporary LVMLs as well as the two open-source LVLMs trained on the training dataset.

However, there are a couple of issues that need to be addressed:
- **Unclear multi-turn scenario in MMDU**: Although the multi-turn aspect of MMDU is emphasized in the paper there is little discussion of the reasoning skills that motivate this aspect. There is a single example of the multi-turn scenario throughout the paper ( in Figure 1) and the only challenge there seems to be the long context and not much in terms of visual reasoning. In my opinion, this was a lost opportunity to create data that requires fine-grained multi-image visual reasoning over a long context or multiple turns.
- **Lack of visual reasoning requirements for ‘solving’ dataset**: The paper barely analyzes the kind of visual reasoning skills required by LVLM models to solve this dataset. This, coupled with the small improvements on existing benchmarks (Table 3) suggests that MMDU/MMDU-45K neither evaluates nor improves the visual reasoning of LVLMs. The dataset seems to focus more on the text aspect and providing long answers that require world knowledge rather than visual reasoning.

**Strengths:**

See Review.

**Additional Feedback:**

- Fig. 2: Data generation pipeline
- Line 213: Tab 4.2 is wrong

**Clarity:**

The paper requires more motivation for the datasets, experimental details for training of the models, and analysis of the visual reasoning skills required for solving the MMDU dataset.

**Documentation:**

Yes

**Limitations:**

Yes

**Opportunities For Improvement:**

In addition to the drawbacks outlined under ‘Review’, the following points are to be considered:
- The paper needs to include a dataset comparison table to highlight the merits of the MMDU benchmark compared to existing benchmarks.
- The data collection process needs to have better motivation, from the aspect of multi-turn and multi-image understanding.
- More analysis is required on the kind of reasoning required to solve this dataset.
- **The term 'dialog' is misleading in the title because this dataset has nothing to do with dialog**.
- Since long-context is another aspect of this dataset that has been highlighted in the paper, it needs experimental results on long-context scenarios to support it (e.g., how does the performance change over number of tokens).

Some questions about experiments:

- Line 65 - Is the SFT stage here the same as instruction tuning stage? How exactly are the models trained? Is the dataset added to existing pool or do you perform continual training on the MMDU-45K dataset?
- Since LLaVA is trained to process a single image only, it is unclear how the MMDU datasets is added to LLaVa.
- Any particular reason why InternVLM benefits more from MMDU than LLaVA?

**Relation To Prior Work:**

Yes

**Summary And Contributions:**

This paper presents a multi-turn, multi-image vision-language benchmark dataset - MMDU - for evaluating the reasoning capabilities of Large Vision Language Models (LVLMs) over multiple images in a multi-turn scenario. Notably, the MMDU dataset has longer tokens and more images than existing benchmarks, potentially posing the problem of vision-language reasoning in longer contexts. The dataset is created by clustering images belong to a broad category and using GPT4o + prompt templates. The benchmark dataset is further validated by experts to create a high-quality benchmark. Evaluation of contemporary open-source and closed-source LVLMs shows that there is significant room for improvement on this benchmark. The paper further introduces MMDU-45K, a multi-turn multi-image training dataset and continues training of two open-source models on this dataset, resulting in large improvements on the MMDU benchmark and smaller improvements on some existing vision-language benchmarks.

---

> ### Author Rebuttal · Authors · 2024-08-16
>
> >Q1: Unclear multi-turn scenario: little discussion of the reasoning skills. Only a single example of the multi-turn scenario ( in Fig. 1). Only has long context challenges. Do not require fine-grained visual reasoning.
>
> This is not true. In addition to Fig. 1, we also presented additional examples in the supplementary material (Fig. 4 and Fig. 5). Our MMDU requires fine-grained multi-image visual reasoning abilities, which are demonstrated in Fig. 4 (architectural structures) and Fig. 5 (types of ships) in the supplementary materials. Thanks to the clustering process during data collection, many of the images in the MMDU dataset are highly related, often sharing fine-grained features. As a result, the MMDU benchmark places high demands on the model's reasoning and comprehension abilities.
>
> >Q2: The paper needs to include a dataset comparison table
>
> In Fig. 1b of our paper, we have already presented a comparison between MMDU and existing datasets. As suggested, we transfer Fig.1b into Tab.1 in the PDF.
>
> >Q3: Data collection process needs better motivation in multi-turn and multi-image understanding.
>
> Thanks for your comment. As mentioned in lines 26-38, our primary motivation is based on human-AI interaction, which is essential for models to comprehend multi-image understanding and multi-turn dialogues within contextual scenarios. Given that most existing benchmarks are focused on short-text, single-image data, we propose MMDU which is beneficial for the community.
> Additionally, current open-source models (e.g., LLaVa) perform poorly on various multi-image benchmarks. However, the model SFT on the MMDU-45k showed significant improvement in results across multiple single-image (Table 3 in the paper) and multi-image benchmarks (Q4-2 below), which reflects the enhancement of reasoning and comprehension ability from single to multiple images.
>
>  >Q4-1: Lack of visual reasoning requirements.
>
> Good question! When answering questions from MMDU, the model must process interleaved image and text inputs while simultaneously understanding the relationships between multiple images which requires various visual reasoning capabilities. We believe that MMDU's visual reasoning capabilities can be reflected in the following aspects：
> - Visual Perception Reasoning: to interpret and infer information from single or multiple images, recognizing relevant details, patterns, and relationships between the images.
> - Visual Relationship Reasoning: to understand and articulate the connections and relationships between different images, determining how they interact or contribute to the overall dialogue.
> - High-Level Semantic Reasoning: to grasp the deeper meanings, symbols, or abstract concepts represented across multiple images.
> - Cross-modal reasoning:  to extract visual information from an image and combine it with textual or conversational cues to infer knowledge or information that goes beyond the content of the image.
>
> These reasoning abilities are also considered when constructing the QA pairs for MMDU. In the attached PDF, we've included cases that illustrate the dataset's visual  reasoning capabilities.
>
> >Q4-2: Small improvements on existing benchmarks (Tab. 3) suggest that neither evaluates nor improves the visual reasoning of LVLMs. Focus more on the text aspect & world knowledge rather than visual reasoning.
>
> We believe the limited improvement claim is inaccurate. The benchmarks used in Tab. 3 are based on single-image and short-form multiple-choice questions, while our MMDU is based on multi-image, multi-turn, and free-form VQA. Considering such a big difference, the model fine-tuned on MMDU still managed to maintain performance, we consider the improvement shown in Tab. 3 to be quite considerable. Furthermore, we provide additional test results on multi-image benchmarks to demonstrate that using MMDU significantly enhances the multi-image reasoning ability of LVLMs (Tab.3 in the PDF).
>
> >Q5: The term 'dialog' is misleading
>
> In the paper, our use of "dialog" is equivalent to "dialogue," referring to multi-turn text-image interactions between humans and the AI (also understood as chat or conversation). Some significant prior work on the vision and dialogue, such as Visual Dialog  (https://arxiv.org/abs/1611.08669) and CLEVR-Dialog (https://arxiv.org/abs/1903.03166). Our MMDU focuses on dialogues involving multiple images and multiple turns.
>
> >Q6: Needs results on long-context scenarios
>
> Good point! We tested the LLaVa-1.5 baseline (context-window length is 4k) and LLaVa-1.5 (SFT on MMDU-45k, we extended the context-window length to 8k with the RoPE interpolation) model with different context lengths. From Tab.2 in the PDF, we observe that: (1) As the context length of the model increases, the performance also improves. (2) Finetuning on MMDU can increase the context window size of the LLaVA model.
>
> >Q7-1: Line 65 - SFT same as instruction tuning?
>
> Yes
>
> >Q7-2: How exactly are the models trained?
>
> The model was trained on 8 A100 GPUs with 1 epoch. The learning rate is 2e-4.
>
> >Q7-3: Added to the existing pool or continual training?
>
> We performed continual training. In addition, we also tried these two options and the differences are minimal (see Tab.4 in the PDF).
>
> >Q8: How was the MMDU dataset added to LLaVa
>
> The LLaVA model can support multi-image inputs with only some simple modifications: we merged the embeddings of multiple images with the text inputs before feeding them into the model. We have released the modified LLaVA with multi-image inputs: https://github.com/Liuziyu77/MMDU/tree/main/LLaVA.
>
> >Q9: Why InternVLM benefits more from MMDU than LLaVA?
>
> As a more recent model, InternLM-Xcomposer2 uses a different LLM backbone, a more powerful architecture, larger pre-training and SFT data, and more advanced training strategies than LLaVa-v1.5. These advantages may give InternLM-Xcomposer2 stronger generalization capabilities.

---

> > ### Author Response · Authors · 2024-08-30
> >
> > Dear Area Chair,
> >
> > I hope this message finds you well.
> > I understand that you must be very busy, but we would greatly appreciate your assistance with a matter regarding our paper.
> >
> > We have provided a detailed rebuttal to reviewer URmW’s comments with the following clarifications:
> >
> > 1.The multi-turn scenario examples that reviewer URmW requested are already included in our supplementary material, along with a dataset comparison table in our attached PDF.
> >
> > 2.We have explained in detail the motivation behind constructing the dataset from the perspective of multi-turn and multi-image understanding.
> >
> > 3.We have analyzed the various reasoning abilities required by the dataset.
> >
> > 4.We have provided a comprehensive description of the model training details.
> >
> > 5.Additionally, we have addressed all other concerns raised by reviewer URmW in our rebuttal.
> >
> > Given the approaching deadline, we would greatly appreciate your help in reminding reviewer URmW to review our response and consider adjusting their feedback accordingly.
> >
> > We truly value your time and efforts in managing this process and are grateful for any help you can provide.
> >
> > Thank you very much for your attention and support!
> >
> > Best regards,
> >
> > Authors

---

> ### Author Response · Authors · 2024-08-26
>
> Dear Reviewer,
>
> We are sorry if this message interrupts you.
>
> We appreciate your comment on our paper. However, we noticed that there are some factual errors (e.g., actually we provided several examples of the multi-turn scenario in the supplementary material) in the comment. And we have clarified accordingly in our rebuttal.
>
> Would you mind checking the rebuttal and, if possible, adjust the score based on our response?
>
> Authors

---

> ### Author Response · Authors · 2024-08-27
>
> Dear Area Chair,
>
> We are sorry if this message interrupts you.
>
> We appreciate reviewer URmW’s contributions to paper reviews for NeurIPS 2024. However, we noticed that there are some factual errors (e.g., actually we provided several examples of the multi-turn scenario in the supplementary material) in the comment. And we have clarified accordingly in our rebuttal.
>
> The deadline for reviewer-author discussion is approaching quickly, but we receive no response from the reviewer URmW. Would you mind reminding URmW to read our rebuttal and post feedbacks with a potential adjustment of the paper rating?
>
> Thanks for your kind help!
>
> Authors

---

> ### Author Response · Authors · 2024-08-31
>
> Dear Reviewer URmW,
>
> Please accept our sincere apologies for the multiple follow-ups. We have not yet received your feedback, and with the review deadline fast approaching (only one day remaining), it is our responsibility to ensure that all your concerns are fully addressed. If you have any further concerns, we would be happy to discuss them with you.
>
> Best regards,
>
> Authors

---

> > ### Comment · Reviewer_URmW · 2024-08-31
> >
> > Dear Authors,
> >
> > Thank you for the detailed rebuttal to each issue raised in my review. Apologies for not being able to engage earlier in this discussion. I am satisfied with most of the arguments presented in the rebuttal except for a few.
> >
> > - Fine-grained visual reasoning: The MMDU dataset does exhibit the need for visual reasoning across images, however, the conversations are generally about prominent objects and visual features in the images. A majority of the textual response is drawn from extensive world knowledge about the featured subjects and objects, which makes it easier for the underlying LLMs to generate a suitable response. However, this may not be a deal-breaking drawback; the main purpose served by this dataset is to improve reasoning across images and it does accomplish this objective to some extent.
> >
> > - Dialog: Yes, the term dialog has been used in a few previous datasets - which were also ill-suited in my opinion. Even so, the nature of dialog in CLEVR-Dialog is closer to human dialog. For instance, the multi-turn questions in CLEVR-Dialog are similar to how one might converse about a product while shopping in real life. The 'dialog' in MMDU is synthetic and comes across as an aptitude test. Hence, my point about 'dialog' not being suitable for the name of this dataset - and I stand by it.
> >
> > Nevertheless, many of my concerns have been addressed, so I will be willing to raise my scores.

---

> ### Author Response · Authors · 2024-08-31
>
> Dear Reviewer URmW,
>
> We are delighted to receive your insightful feedback and are grateful for your willingness to consider improving our score.
> Regarding the two new issues you mentioned, we have provided our responses below:
>
> 1. Fine-grained Visual Reasoning: While it is true that generating textual responses requires extensive world knowledge, it is also essential for the model to accurately understand and perceive the content of the images. Without a correct visual understanding, the model cannot effectively apply its world knowledge. This is especially critical in multi-image scenarios, where a lack of visual reasoning capabilities across multiple images can lead to significant errors, such as confusing elements from different images (e.g., misinterpreting the main subject of the second image as being in the first image). In our dataset, we have taken measures to avoid situations where a model could generate textual responses solely based on its world knowledge without comprehending the content of multiple images.
>
> 2. Dialog: We acknowledge the validity of your points regarding dialog. As our dialog data is synthetically generated, it indeed differs from natural human dialog. However, compared to existing datasets used for training large vision-language models (LVLMs), such as those based on multiple-choice questions and caption data, we have strived to ensure that both the textual content and the number of images more closely resemble real human dialogs. In future work, we plan to enhance the human-like quality of dialogs by incorporating role prompts and using role-playing methods.
>
> Lastly, we sincerely thank you for taking the time to review our rebuttal and for your thoughtful consideration. We greatly appreciate your attention and feedback.
>
> Best regards,
>
> The Authors

---

### Official Review · Reviewer_boEu · 2024-08-09
**MMDU: A Multi-Turn Multi-Image Dialog Understanding Benchmark**

**Rating:** 7
**Confidence:** 4

**Review:**

The submission presents a well-constructed and open-sourced benchmark for Multi-Turn Multi-Image Dialog Understanding (MMDU). The primary novelty lies in the innovative pipeline that leverages single items to create a comprehensive MMDU dataset. This approach showcases significant potential in advancing the field of dialog understanding, particularly in multi-turn, multi-image contexts.

However, there are notable concerns regarding the methodology. The process of combining multiple items based on potentially inaccurate clustering could introduce noise and errors into the dataset, potentially affecting the overall quality and reliability of the benchmark. Additionally, the evaluation pipeline for MMDU seems somewhat limited, as GPT-4 score (or GPT-4o score here) appears to focus more on assessing overall language quality rather than detecting subtle factual errors.

**Strengths:**

1. The dataset addresses a critical gap in Multi-Turn Multi-Image Dialog Understanding.

2. A novel data preparation pipeline that uses single items to construct a MMDU dataset.

3. A pipeline that employs a smaller language model as a filter and utilizes a multimodal language model like GPT-4o for dataset construction.

**Additional Feedback:**

None.

**Clarity:**

The paper is well-organized and clearly written, making it easy to follow the authors' arguments. However, a few minor improvements could enhance clarity:

In Figure 2(a), clusters 1 and 2 should be swapped for consistency.

In Figure 5(a), it's unclear which images correspond to 1, 2, 3, and 4. While this information is available in the caption, updating the image itself for better clarity would be helpful.

**Correctness:**

The methods for data collection and labeling appear sound and are supported by comprehensive documentation.

**Documentation:**

The documentation is comprehensive, with detailed information on data collection processes, intended uses, and ethical considerations.

**Ethics:**

No significant ethical concerns were identified, although further clarification on data consent processes would be beneficial.

**Limitations:**

The author states the limitations of this paper and adds ethical considerations.

**Opportunities For Improvement:**

1. The process of combining multiple items based on potentially inaccurate clustering could introduce noise and errors into the dataset, potentially affecting the overall quality and reliability of the benchmark. For example, the "Starry Night" cluster should include other significant works by Vincent van Gogh, like "Sunflowers," and should avoid including unrelated pieces such as a Chinese landscape painting. A possible solution could be to use a language model like GPT-4 to verify the clusters. This is my primary concern, and I may reconsider my score based on your response to this issue.

2. The Human Check process is not well discussed. How many of the 45k samples were verified by humans? The paper mentions that "the first round involved preliminary checks by regular reviewers, and the second round involved in-depth examination and modification by experts." However, it's unclear whether the human reviewers manually wrote the ground truth answers or simply provided scores. This needs to be clarified.

3. The evaluation using GPT-4 appears limited and may not fully capture the nuances necessary for robust assessment.

**Relation To Prior Work:**

The authors provide a thorough comparison with existing datasets and clearly articulate the unique contributions of their work. However, I have a question: Is this the first paper to use the idea of constructing a multi-turn and multi-image dataset from single items?

**Summary And Contributions:**

The authors have developed a new benchmark for Multi-Turn Multi-Image Dialog Understanding (MMDU), offering several key contributions:

(1) A novel data collection pipeline that utilizes single items to generate multi-turn, multi-image dialog scenarios.

(2) A dataset construction method that employs a small language model as a filter and leverages a multimodal large language model for more refined dataset creation.

(3) An evaluation approach that incorporates GPT-4o for MMDU.

---

> ### Author Rebuttal · Authors · 2024-08-16
>
> Dear reviewer,
>
> We sincerely thank you for your thorough feedback: a novel data collection pipeline and a well-constructed and open-sourced benchmark. We address the questions below and will incorporate all feedback in the final version.
>
> > Q1: The process of combining multiple items based on potentially inaccurate clustering could introduce noise and errors into the dataset, potentially affecting the overall quality and reliability of the benchmark. For example, the "Starry Night" cluster should include other significant works by Vincent van Gogh, like "Sunflowers," and should avoid including unrelated pieces such as a Chinese landscape painting. A possible solution could be to use a language model like GPT-4 to verify the clusters. This is my primary concern, and I may reconsider my score based on your response to this issue.
>
> Great suggestion! In the process of constructing multi-item clusters, we ensure clustering accuracy through two key steps: 1. We first utilize the inherent tags or labels associated with each wiki item for clustering. These tags and labels are manually annotated in wiki items, providing a high level of accuracy. Additionally, we further employ image captions for clustering, ensuring that the resulting clusters are largely free of noise and errors.
>
> Considering your proposal to use GPT-4 for verifying the clusters, we conducted the following experiment: We randomly sampled five sets of data from the MMDU-45k dataset, each containing 100 entries. The images from each entry were input into the GPT-4o model, which was tasked with evaluating whether the multiple images were related and could be grouped into the same cluster. The resulting accuracy rates were recorded as follows: 94%, 90%, 92%, 89%, 91%. This indicates that the clustering accuracy of MMDU is very high. The prompt we used is included in the attached PDF.
>
> Regarding the case you mentioned in Figure 1, where "Starry Night" and a Chinese painting were clustered together, we believe they both belong to the category of artworks. Furthermore, on this basis, GPT-4o can easily generate a logically coherent multi-turn dialogue using these two images. We have included dialogue examples based on these two images in the attached PDF.
>
>
> > Q2: The Human Check process is not well discussed. How many of the 45k samples were verified by humans? The paper mentions that "the first round involved preliminary checks by regular reviewers, and the second round involved in-depth examination and modification by experts." However, it's unclear whether the human reviewers manually wrote the ground truth answers or simply provided scores. This needs to be clarified.
>
> Valuable feedback! For the dialogues in the MMDU benchmark, our experts reviewed and corrected (by removing or rewriting) any hallucinations and errors to ensure that all dialogues are accurate. Due to the large volume of data in the MMDU-45k, it was not feasible to review all of the data, so we sampled 5% of the dataset for inspection. Statistical analysis showed that the probability of hallucinations and errors in this subset was less than 5%, indicating a high level of reliability. To facilitate verification, we also designed a specialized web UI that allows for quick browsing and modification of data content. Please refer to the PDF in the General Response section to check the interface of our web UI used for the human check process. We will update the details of the human check process in our revised version.
>
>
> > Q3: The evaluation using GPT-4 appears limited and may not fully capture the nuances necessary for robust assessment.
>
> Good question! Evaluating the subjective, open-ended,  free-form, and long-context visual question-answering pairs is indeed challenging. In general, the community considers two options:
>
> (1) Traditional metrics (e.g., BLEU-4, CIDEr). Since these metrics often suffer from shortcomings like neglecting semantic understanding and failing to capture long-distance dependencies, they are not popular choices in recent days.
>
> (2) LLM/LVLM models as a judge. This method excels at understanding context and semantics, providing more accurate evaluations of visual content, and capturing long-context dependencies, which traditional metrics often miss.
>
> While using LLM/LVLM models like GPT-4o as a judge indeed has certain limitations (e.g., sensitivity to evaluation prompts), they are increasingly recognized as better alternatives for evaluation. Recent LLM benchmarks (e.g., MT-Bench [1]) and Multi-modal benchmarks (e.g., MM-Vet [2]) also use the LLM/LVLM for judging VQA/dialog understanding tasks.
>
> As for your concern that ''GPT-4 may not fully capture the nuances necessary'', we have meticulously crafted evaluation prompts to let the GPT consider the nuances. Besides, we also present the following two proofs to demonstrate the effectiveness of GPT-4o judgment:
>
> - In line 202 of the main paper, we compared the assessment results of GPT-4o with the human experts. The GPT-4o exhibits Pearson, Spearman, and Kendall similarities of 97.5%, 97.3%, and 89.0% respectively, with human judgment results.
>
> - In Table 1 and Figure 8 of the Supplementary Material, we compare the difference of using different judgment models (e.g., GPT-4o, GPT-4-turbo, Claude 3-Opus). We find a strong similarity between the evaluation outcomes of different models, and overall the GPT-4o model has the the maximum correlations with the human experts.
>
> Based on the above-mentioned reasons, we select the GPT-4o as the judgment model.
>
> [1] Judging LLM-as-a-Judge with MT-Bench and Chatbot Arena, Neurips 2023 Datasets and Benchmarks Track
>
> [2] MM-Vet: Evaluating Large Multimodal Models for Integrated Capabilities, ICML 2024

---

> ### Author Response · Authors · 2024-08-27
>
> Dear Reviewer boEu,
>
> We sincerely thank you again for your great efforts in reviewing this paper, especially for the precious advice that has helped us improve the quality of this paper significantly!
>
> In response to your suggestions, you raised concerns that combining multiple items based on potentially inaccurate clustering might introduce noise and errors into the dataset. To address this, we conducted additional experiments, using GPT-4o to validate the clustering accuracy, and demonstrated the reliability of this method. Regarding your point about clustering Van Gogh's Starry Night with Chinese-style paintings, we consider both to be part of the broader category of artworks, with dialogue examples generated by GPT-4o provided in the attached PDF.
>
> We provided further details on the human check process in Q2 of the rebuttal. As for your concerns about using GPT-4o for scoring, we have explained the reasons for this approach in Q3, comparing its strengths and weaknesses with other subjective evaluation methods.
>
> Would you mind checking the rebuttal and, if possible, adjust the score based on our response? Thanks for your kind help!
>
> Best,
>
> Authors.

---

> > ### Comment · Reviewer_boEu · 2024-08-30
> > **Response**
> >
> > Dear Authors,
> >
> > Thank you for your detailed and thoughtful response to my review. I appreciate the additional experiments you conducted using GPT-4o to validate the clustering accuracy and the clarifications you provided regarding the human check process and scoring methodology. Given this, I am inclined to adjust my score in light of the improvements and clarifications you’ve made.

---

> ### Author Response · Authors · 2024-08-31
>
> Dear Reviewer boEu,
>
> Thanks for your positive feedback on our work!
>
> I'm glad to hear that our rebuttal has addressed your concerns regarding (1) cluster accuracy, (2) the human check process, and (3) the scoring methodology. We sincerely appreciate your time and effort in providing such meticulous reviews and insightful comments.
>
> Best regards,
>
> Authors

---

### Author Rebuttal · Authors · 2024-08-16

Dear Reviewers,

Thank you for taking the time to review our manuscript and for providing thoughtful and constructive feedback. We are delighted to see that the reviewer acknowledged the novel data collection pipeline (boEu), high-quality benchmark (URmW), comprehensive evaluation (Ckq6), and well-organized and clearly written (boEu).
Below we summarize some major points we have addressed in the rebuttal. More detailed responses are provided individually for each reviewer.

**More information on data collection based on clustering.** We have further clarified the process of using clustering methods to collect data and have validated the accuracy of this approach with GPT-4o (Reviewer boEu).

**The details of the human verification process.** We have provided a thorough explanation of the manual verification and modification process of MMDU (Reviewer boEu), as well as the selection and training process of annotators (Reviewer Ckq6). The web UI interface specifically designed for data inspection and modification is included in the attached PDF.

**GPT-4o as the evaluation model.** We have compared traditional methods with current mainstream approaches for evaluating free-form benchmarks and analyzed the reasons for using the GPT-4o model for scoring (Reviewer boEu, Ckq6).

**More details and experiments about the dataset and model training.** This includes further insights into the motivation for dataset collection, an analysis of the dataset’s reasoning capabilities, and the specifics of the model training process. We also tested the model's performance across different context lengths (Reviewer URmW).

**Clarifications on some confusing points.** We have provided further clarification on the SFT stage, the implementation of multi-image LLaVa, the use of "dialog" in the paper (Reviewer URmW), and the issue regarding the number of Q&A pairs in the MMDU benchmark (Reviewer Ckq6).

In the point-to-point responses below, we have provided more detailed responses to each of the reviewers' concerns. Please don't hesitate to let us know if there are any additional clarifications or experiments that we can offer！

Yours Sincerely,

Authors

---

### Decision · Program_Chairs · 2024-09-26

**Decision:**

Accept (Poster)

**Comment:**

The paper introduces MMDU,  a comprehensive benchmark for multi-turn multi-image dialog understanding.  It is 5 times longer than previous benchmarks and has a few nice properties,  e.g., open-ended evaluation to test the performance of LLMs through free-form multi-turn outputs.  The authors have benchmarked many LLMs on this dataset and carefully answered reviewers' questions.  Please revise the paper according to these feedbacks.